# Evaluation of subclinical ventricular systolic dysfunction assessed using global longitudinal strain in liver cirrhosis: A systematic review, meta-analysis, and meta-regression

**Denio A. Ridjab**[1☺]*, **Ignatius Ivan**[2☺], **Fanny Budiman**[2], **Riki Tenggara**[3]

1 Department of Medical Education Unit, School of Medicine and Health Sciences, Atma Jaya Catholic University of Indonesia, Jakarta, Indonesia, 2 Fifth Year Medical Student, School of Medicine and Health Sciences, Atma Jaya Catholic University of Indonesia, Jakarta, Indonesia, 3 Department of Internal Medicine, School of Medicine and Health Sciences, Atma Jaya Catholic University of Indonesia, Jakarta, Indonesia

☺ These authors contributed equally to this work.
* denio.ridjab@atmajaya.ac.id

**Data Availability Statement:** All relevant data are within the article and its Supporting Information files.

## Abstract

Global longitudinal strain (GLS) can identify subclinical myocardial dysfunction in patients with cirrhosis. This systematic review aims to provide evidence of a possible difference in GLS values between patients with cirrhosis and patients without cirrhosis. Studies from inception to August 11, 2021, were screened and included based on the inclusion criteria. The Newcastle Ottawa Scale was used to assess the quality of nonrandomized studies. Meta-analyses were conducted with subsequent sensitivity and subgroup analyses according to age, sex, cirrhosis etiology, and severity. Publication bias was evaluated using Begg's funnel plot, Egger's test, and rank correlation test with subsequent trim-and-fill analysis. The systematic database search yielded 20 eligible studies. Random effect showed a significant reduction of left ventricular (LV) GLS (MD:-1.43;95%; 95%CI,-2.79 to -0.07; p = 0.04; $I^2$ = 95% p<0.00001) and right ventricular (RV) GLS (MD:-1.95; 95%CI,-3.86 to -0.05, p = 0.04; $I^2$ = 90%, p<0.00001) in the group with cirrhosis. A sensitivity test on subgroup analysis based on the study design showed a -1.78% lower LV-GLS in the group with cirrhosis ($I^2$ = 70%, p = 0.0003). Meta-regression analysis showed that the severity of cirrhosis was significantly related to GLS reduction. This research received no specific grants from any funding agency in the public, commercial, or not-for-profit sectors. The study protocol was registered at PROSPERO (CRD42020201630). We followed the Preferred Reporting Items for Systematic Reviews and Meta-Analyses (PRISMA) 2020 statement guidelines.

## Introduction

Cirrhotic cardiomyopathy (CCM) is defined by the World Congress of Gastroenterology [1], revised in 2019 according to criteria from the Cirrhotic Cardiomyopathy Consortium, as a left ventricular ejection fraction (LVEF) less than or equal to 50% and/or absolute global

**Funding:** The author(s) received no specific funding for this work.

**Competing interests:** The authors have declared that no competing interests exist.

**Abbreviations:** CCM, Cirrhotic cardiomyopathy; EF, Ejection Fraction; STE, Speckle tracking echocardiography; CMR, cardiovascular magnetic resonance; GLS, global longitudinal strain; PROSPERO, Prospective Register of Systematic Reviews; PRISMA, Preferred Reporting Items for Systematic Reviews and Meta-analyses; REM, random-effects model; NOS, Newcastle Ottawa Scale; LV, Left Ventricular; RV, Right Ventricular.

longitudinal strain (GLS) less than 18% in patients with cirrhosis [2]. A hyperdynamic state, characterized by increased cardiac output and decreased peripheral vascular resistance, occurs in patients with liver cirrhosis [3]. This hyperdynamic condition can progress to abnormal contractile response to stress and/or altered diastolic relaxation, with electrophysiological abnormalities in the absence of known cardiac disease [4–6]. Myocardial stiffness and subsequent hypertrophy due to sodium and fluid retention can also lead to diastolic dysfunction [7,8]. CCM is often under-recognized, although it is acknowledged that early recognition is important in determining prognosis, especially for patients who may require future procedures that could potentially stress the heart, such as shunt implantation and liver transplantation [9].

Many modalities are used to measure myocardial strain, or regional deformation of the myocardium, and evaluate myocardial performance [5,10,11]. Myocardial strain measurements can be a better quantitative approach compared with the conventional ejection fraction (EF) measurement, which is highly dependent on geometrical assumptions and endocardial border definition [11,12]. EF evaluation is limited by reader experience and does not consider that a hyperkinetic segment may compensate for a hypokinetic one, leading to a false "normal" result [12]. These limitations decrease the sensitivity of EF to diagnose myocardial dysfunction at an early stage [11]. Speckle tracking echocardiography (STE) has become the preferred modality to measure myocardial based on the change in myocardial length in one region [11] Myocardial strain can be classified into circumferential, longitudinal, radial, and transverse strains; these are better at evaluating contractile function than left ventricular ejection fraction (LVEF) alone, which primarily reflects radial function [2]. Global longitudinal strain (GLS) analysis using STE has demonstrated the ability to identify subclinical myocardial dysfunction in various diseases [13]. The use of GLS to identify myocardial contractile dysfunction in patients with preserved LVEF is important because longitudinal contractile function is often impaired before radial contractile function [2]. Other modalities, such as cardiovascular magnetic resonance (CMR), have also been utilized as a standard reference for evaluating cardiac morphology, volume, and myocardial strain [10]. According to the latest CCM consortium [2], data on strain imaging to detect CCM in patients with normal LVEF are limited and conflicting, with three studies showing normal longitudinal strain [10,14,15] and one multi-center study showing diminished longitudinal strain in one of two cohorts of patients with cirrhosis [16]. Thus, our systematic review aimed to evaluate whether GLS values differ between patients with cirrhosis and patients without cirrhosis.

## Materials and methods

The study protocol was registered at the International Prospective Register of Systematic Reviews (PROSPERO) with registration number CRD42020201630 [17]. This systematic review and meta-analysis followed the Preferred Reporting Items for Systematic reviews and Meta-Analyses (PRISMA) 2020 guidelines for reporting [18]. Furthermore, we complied with the guidelines for conducting systematic reviews and meta-analyses of observational studies of etiology (COSMOS-E) [19]. We used a pre-determined search strategy to conduct a structured search of the literature to identify studies on the outcome of GLS in patients with cirrhosis versus patients without cirrhosis. The search was conducted in the Cochrane Library, EBSCOhost, Open Grey, PMC, ProQuest, PubMed, and ScienceDirect databases from inception to August 11, 2021; additionaly, a manual search was performed to retrieve relevant studies. Using MeSH terms and [All Field], we complemented the search strategy using the following keywords: ventricular function, ventricular dysfunction, myocardial, ultrasonography, echocardiography, cardiac magnetic resonance, speckle tracking, longitudinal strain, tissue-Doppler, liver cirrhosis, and end-stage liver disease.

The results of the search strategy were exported to Endnote X9, duplicates were removed, and the remaining articles were reviewed based on the title and abstract. Studies were included based on the following criteria: (1) observational studies with participants aged > 18 years; (2) article written in English; and (3) availability of GLS data from patients without cirrhosis versus patients with cirrhosis estimated using mean or median. Studies were excluded if: (1) study participants aged > 80 years; (2) participants with reduced/mid-range EF heart failure as defined by the European Society of Cardiology 2016 guideliens [20]; (3) no control groups available for complete data extraction; (4) data for ventricular longitudinal strain not available; and (5) full articles not retrieved. After selection of all studies fulfilling the inclusion criteria, the following data were extracted: first author, publication year, country of origin, sample size, age of participants, and GLS values. For duplicate data extraction, two authors performed data extraction to reduce the possibility of a single person's bias. The study authors were contacted via email to request access to missing data. The mean and median estimations of GLS were assessed to elucidate possible differences between patients with and without cirrhosis. The mean difference and 95% confidence interval (CI) were used to determine the difference between the compared groups. Thus, all studies reporting median estimations were approximated into mean estimations using the method proposed by Wan et al., which performs very well for both normal and skewed data [21].

To detect statistical heterogeneity, we used Cochrane's Q test (chi-squared test) and Higgins $I^2$ statistics. Heterogeneity was considered to be present if $P < 0.10$ or $I^2 > 75\%$ [22,23]. A forest plot was generated to evaluate heterogeneity. If heterogeneity was present, we performed a random-effects model (REM) using the DerSimonian-Laird method [24].

Publication bias analyses were performed using JASP version 0.16.1 (JASP Team, Amsterdam) [25]. Begg's funnel plot was generated to assess publication bias when the number of included studies was at least 10 and heterogeneity was not statistically significant [22,26]. This was further confirmed by Egger's test and Begg and Mazumdar's rank correlation test [27,28]. Correction of publication bias was based on Duval and Tweedie's trim-and-fill method if the heterogeneity was less than 75% [26,29]. The overall fail-safe number of publications to assure a borderline significant effect size was calculated according to Rosenthal et al [30]. When the fail-safe number was relatively large compared to the number of included studies, higher confidence was assured regarding the stability of the results. A fail-safe number is considered robust if it is five times higher than the number of studies included, plus 10 [30].

We performed a random-effects (method of moments) meta-regression using Comprehensive Meta-Analysis version 3.3.070. software (Biostat Inc., Englewood, NJ, USA) [31]. Meta-regression was conducted to investigate the true causes of heterogeneity that explained the high value of the $I^2$ statistic [32]. Sources of potential variability were based on covariates of study design, proportion of male subjects, mean age of study sample, mean model for end-stage liver disease (MELD) score, proportion of decompensated cirrhosis (Child-Pugh class B and C), proportion of patients with alcoholic-etiology cirrhosis, proportion of patients with viral-etiology cirrhosis, baseline LVEF in the group with cirrhosis, and methodological quality of the study.

For sensitivity analysis, we deleted one study at a time to determine the effect and stability of one study on the pooled mean difference [33]. Subgroup analysis was performed when at least 10 studies were included [33]. Subgroup analysis was performed based on the study design, despite no strong evidence of statistical heterogeneity [34]. Subgroup analysis was also performed based on age, sex, cirrhosis etiology, and severity (MELD or Child-Pugh classification) to detect clinical heterogeneity. These variables were chosen because previous studies support the potential effects of age [2,35], sex [2], cirrhosis etiology [36], and severity [2,37–40] on GLS results. For cirrhosis severity, if provided by at least 10 studies [33], subgroup

analysis was performed by dividing patients with cirrhosis into compensated (Child-Pugh class A) and decompensated (Child-Pugh class B and C) groups as classified in a previous study [36]. Furthermore, sensitivity analysis was also performed within each subgroup analysis to reduce heterogeneity to less than 75% by omitting the study with the lowest methodological quality while also having the largest heterogeneity contribution. Meta-analysis was performed using Review Manager 5.3.5 software (Copenhagen: The Nordic Cochrane Centre, The Cochrane Collaboration) [41].

The methodological quality of observational studies was in accordance with the Newcastle Ottawa Scale (NOS) [42]. The NOS assesses participant selection, comparability, and outcome reporting using eight subscale items [42]. For cross-sectional studies, an adapted version of NOS was used, similar to previous studies [43,44]. The sum of the subscale item scores, with a maximum of 10, was used to provide an overall assessment of evidence quality for each study. For case-control and cohort studies, a maximum score of nine from the sum of the subscale items was used [42]. All risk of bias analyses were performed by two authors, and disagreements were resolved by a third author.

## Results

The search strategy identified 5347 studies from the database searches (S1 Table), and the manual search identified seven additional studies. The results were imported into Endnote X9, and duplicates were removed, leaving 5068 articles for review. The article abstracts were reviewed for relevance based on inclusion and exclusion criteria. After screening, 26 studies from the databases and seven studies from manual searching methods were retained for full review. Of these, 20 were retained for the analysis (Fig 1). The other 13 studies were excluded

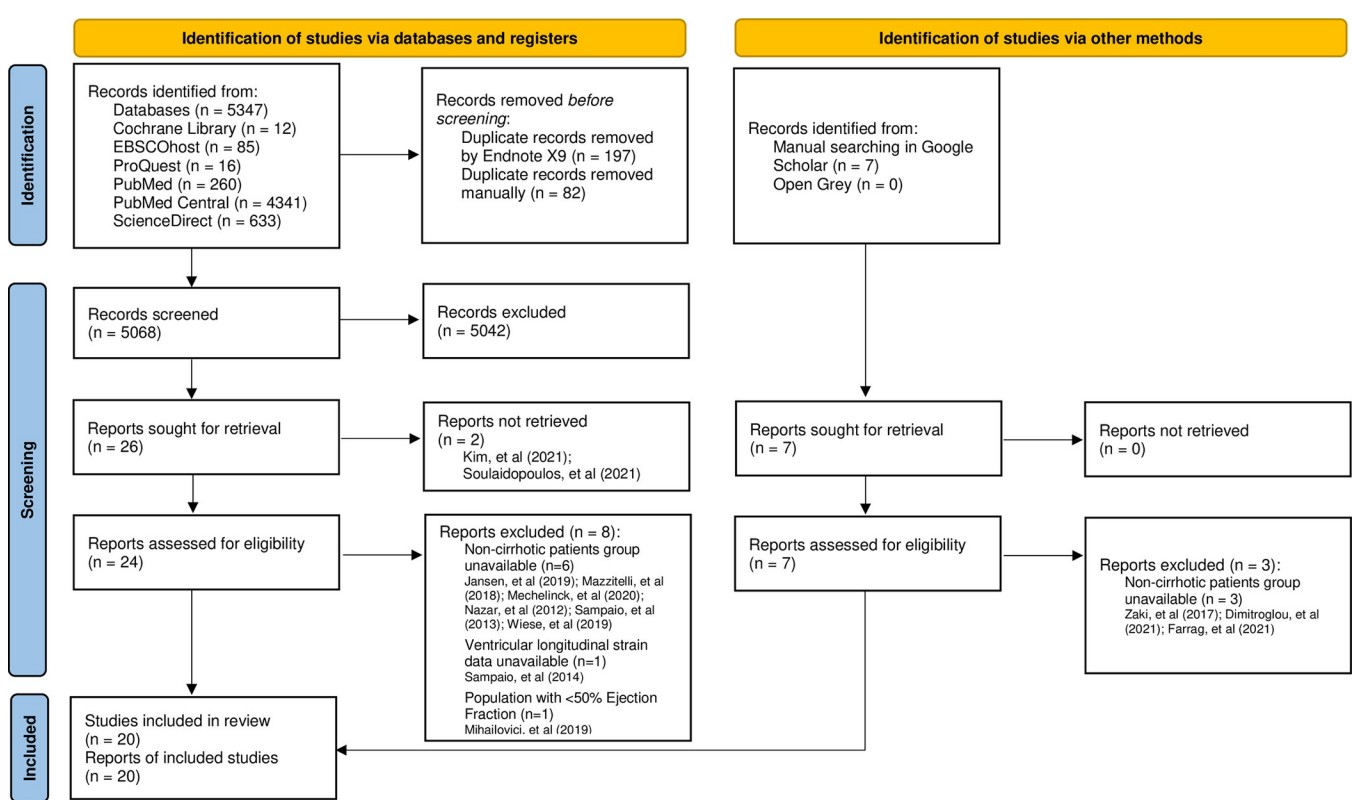

**Fig 1. PRISMA 2020 flow diagram of the identification and selection of studies included in the analysis.**

for the following reasons: full-text failed to be retrieved in two studies, comparison with patients without cirrhosis was unavailable in nine studies, ventricular longitudianl strain was not analyzed in one study, and population with EF < 50% in one study.

This systematic review identified six cross-sectional [11,15,45–48], 6 case-control [10,49–53], and eight prospective cohort studies[14,36,54–59] gathered from 12 countries and consisting of 1738 participants (Table 1). Our search strategy did not identify any randomized controlled trials. We found one cross-sectional [15], one case-control [53], and three prospective cohort studies [14,55,59] providing results for GLS in both ventricles. There were four cross-sectional [11,45,46,48], five case-control [10,49–52], five prospective cohort studies [36,54,56–58] providing results only for the left ventricle. Meanwhile, one cross-sectional study [47] providing result only for the right ventricle. There were two case-control studies [10,52] in which evaluations were performed using CMR and others using 2D-STE. Five cross-sectional [11,15,45,47,48], three case-control [50,51,53], and five prospective cohort studies [14,36,54,57–59] used the EchoPAC system; one cross-sectional [46], one case-control [49], and one prospective cohort study [55] used Velocity Vector Imaging and one prospective cohort study [56] used the QLAB system for 2D-STE evaluation. The etiology of cirrhosis varied; the most common etiologies were hepatitis B and C viruses [10,14,15,36,45,46,48,49,52–54,56–59]. Three studies did not specify the etiology [11,51,55]. Comorbidities outside cardiac pathology among the cirrhotic group included hypertension and diabetes mellitus, which were reported in one cross-sectional [46] and two prospective cohort [14,57] studies. Other comorbidities such as dyslipidemia and pulmonary artery hypertension were reported in one cross-sectional [46] and one case-control study [51], respectively.

All five prospective cohorts had different durations of follow-up, including 7 [54], 8[56], 12 [36], 14 [58], 19 [59], 22 [57], and 32 months [14]. One study [55] did not report the follow-up duration. Altekin et al. [54] included patients with cirrhosis between the ages of 20 and 65 years, with 60.5% in the compensated group and 39.5% in the decompensated group. The most common etiology was viral hepatitis B, with a mean MELD score of 11.76. Huang et al. [36] included patients with cirrhosis between the ages of 35 and 65 years, with 38.75% in the compensated group and 61.25% in the decompensated group. The most common etiology was viral hepatitis B and C, with mean MELD score of 15.47. Inci et al. [55] included patients with cirrhosis between the ages of 30 and 60 years, with all subjects in the decompensated group. In this study, the etiology and mean MELD scores were not reported. Özdemir et al. [56] included patients with cirrhosis aged 30–60 years. The proportion of patients with compensated or decompensated cirrhosis has not yet been reported. All cirrhosis etiologies were due to hepatitis B, and the mean MELD score was not reported. Kim et al. [57] included patients with cirrhosis between the ages of 40 and 70 years, with 30.3% in Child-Pugh A and B, and 69.7% in Child-Pugh C. The most common etiology was hepatitis B, with a mean MELD score of 18.8. Chen et al. [14] included patients with cirrhosis between the ages of 40 and 70 years, with 71.64% in the decompensated group. The most common etiology was viral infection, with a mean MELD score of 17.77. Hassan et al. [58] included patients with cirrhosis between the ages of 20 and 65 years, with 66.66% in the decompensated group. All cirrhosis etiologies were hepatitis C. Mean of MELD score was not reported. Ibrahim et al. [59] included patients with cirrhosis between the ages of 30 and 70 years, with 24% in the decompensated group. All cirrhosis etiologies were hepatitis C. The mean MELD score was not reported.

For left ventricular assessment, there were two cross-sectional [45,46], four case-control [49–52], and four prospective cohort studies [14,54–56] showing a significantly lower absolute GLS in patients with cirrhosis than in patients without cirrhosis. Meanwhile, two cross-sectional studies [11,48] and one prospective cohort study [57] reported a higher absolute GLS in patients with cirrhosis than in patients without cirrhosis. We found one cross-sectional study

**Table 1. Summary of included studies with left and right ventricular global longitudinal strain results.**

| Author (Year) | Subjects (LC [%male] vs non-LC [% male]) | Baseline Ages (LC vs non-LC, years old) [Median, Q1-Q3 / Mean ± SD]) | Cirrhosis Etiology n (%) | Cirrhosis Prognostic Score and Severity | Cirrhosis Group Comorbidity n (%) | Left Ventricular Longitudinal Strain (LC vs non-LC, %) (Median, Q1-Q3 / Mean ± SD) | Right Ventricular Longitudinal Strain (LC vs non-LC, %) [Median, Q1-Q3 / Mean ± SD] | Other Parameter (TAPSE [mm], RVFAC [%]) (LC vs non-LC) |
|---|---|---|---|---|---|---|---|---|
| **CROSS-SECTIONAL** | | | | | | | | |
| Hammami R (2017) [45] | 80 [52.5] vs 80 [N.R] | 55 ± 14 vs 51 ± 12 (p>0.05) | Viral (Hepatitis B and C): 42 (52.6%) Cryptogenic: 21 (26.1) Other causes: 17 (21.25) | Mean MELD Score: 14.2 ± 4.98 Mean CP Score: N.R CP classification n (%): A: 24 (30) B: 36 (45) C: 20 (25) | Left ventricular hypertrophy: 39 (48.75) | Apical 4,2,3-chamber view: -19.8 ± 2.8 vs -22.01 ± 2.6 (p<0.001) | N.R | N.R |
| Rimbaş RC (2017) [15] | 46 (65.2) vs 46 (72.5) | 57 ± 9 vs 55 ± 10 (p>0.05) | Alcoholic = 52% Viral = 41% Primary biliary cirrhosis = 4% Cryptogenic = 2% | Mean of MELD Score: 13 ± 5 Mean of CP Score: 7 ± 2 CP classification (n,%): A: 23 (50) B: 16 (35) C: 7 (15) | Diastolic dysfunction: 22 (47.8) | Apical 4,2,3-chamber and short axis view at level of papillary muscle: -20.9 ± 3 vs -20.7 ± 2.8 (p>0.05) | Apical 4-chamber view: -24 ± 5 vs -23 ± 4 (p>0.05) | TAPSE: 26 ± 5 vs 25 ± 3 (p>0.05) RVFAC: 43 ± 12 vs 44 ± 8 (p>0.05) |
| Novo G (2018) [46] | 39 [41.02] vs 39 [43.58] | 60 (48-70) vs 59 (48-67) (p>0.05) | Hepatitis C: 39 (100) | Median of MELD score: 7 (6–8) Median of CP score: N.R CP classification (n,%): A: 39 (100) | Hypertension: 17 (43.58) Diabete mellitus: 11 (28.20) Dyslipidemia: 2 (5.13) | Apical 4,2,3-chamber view: Median: -18.1 (-20.5 to -16.3) vs -21.2 (-22.3 to -20.4) (p = 0.001) Mean estimation (Wan's method): -18.3 ± 3.23 vs -21.3 ± 1.46 (p<0.001) | N.R | TAPSE: 22 (21-24) vs 23 (20-25) (p = 0.702) Mean Estimation (Wan's method): 22.33 ± 2.31 vs 22.67 ± 3.85 (p>0.05) |
| Zamirian M (2019) [11] | 20 [50] vs 10 [80] | 42.2 ± 4.7 vs 41.6 ± 4.7 (p>0.05) | N.R | Mean of MELD score: N.R Mean of CP score: N.R CP classification (n,%): C: 20 (100) | Diastolic dysfunction: 4 (20) | Apical 4,2,3-chamber view: -22.6 ± 2.4 vs -19.2 ± 1.9 (p = 0.001) | N.R | N.R |
| Zhang K (2019) [47] | 67 [44] vs 36 [49] | 53 ± 12 vs 47 ± 13 (p>0.05) | Alcoholic: 67 (100) | Mean of MELD score: 13.5 ± 8.9 Mean of CP score: N.R CP classification (n,%): N.R | N.R | N.R | Apical 4- chamber view: -19.8 ± 4.2 vs -21.8 ± 1.7 (p = 0.005) | TAPSE: 25 ± 5 vs 25 ± 4 (p = 0.809) RVFAC: 48 ± 10 vs 47 ± 8 (p = 0.483) |
| von Köckritz F (2021) [48] | 80 [58.8] vs 30 [46.7] | 52.47 ± 10.24 vs 48.57 ± 12.93 (p = 0.145) | Alcoholic: 31.25% Hepatitis C: 12.5% Autoimmune: 10% NASH: 10% PSC: 8.75%, Idiopathic: 8.75% Other cause (cystic liver, Wilson's disease, bile duct carcinoma, and Caroli syndrome): 18.75% | Mean of MELD score: 17 ± 6.65 Mean of CP score: N.R CP classification (n,%): C: 80 (100) | Diastolic dysfunction: 14 (17.5) | Apical 4,2,3-chamber view: -21.39 ± 4.06 vs -18.73 ± 2.95 (p<0.001) | N.R | N.R |
| **CASE-CONTROL** | | | | | | | | |

*(Continued)*

**Table 1.** (Continued)

| Author (Year) | Subjects (LC [%male] vs non-LC [% male]) | Baseline Ages (LC vs non-LC, years old) [Median, Q1-Q3 / Mean ± SD]) | Cirrhosis Etiology n (%) | Cirrhosis Prognostic Score and Severity | Cirrhosis Group Comorbidity n (%) | Left Ventricular Longitudinal Strain (LC vs non-LC, %) (Median, Q1-Q3 / Mean ± SD) | Right Ventricular Longitudinal Strain (LC vs non-LC, %) [Median, Q1-Q3 / Mean ± SD] | Other Parameter (TAPSE [mm], RVFAC [%]) (LC vs non-LC) |
|---|---|---|---|---|---|---|---|---|
| **CROSS-SECTIONAL** | | | | | | | | |
| Sampaio F (2013) [49] | 109 [78.9] vs 18 [17.2] | 54 (48–64) vs 51 (49–58) (p>0.05) | Alcoholic: 73 (67) Viral: 27 (24.8) Other: 9 (8.2) | Median of MELD score: 14 (10–18) Median of CP Score: N.R CP classification (n,%): A: 37 (33.9) B: 27 (24.8) C: 45 (41.3) | Diastolic dysfunctoin: 44 (40.3) | Apical 4,2-chamber view: -19.99% (-21.88 to -18.71) vs -22.02% (-23.10 to -21.18) (p = 0.003) Mean estimation (Wan's method): -20.19 ± 2.38 vs -22.10 ± 1.54 (p<0.05) | N.R | TAPSE: 25.4 mm (22.0–28.2) vs 23.1 (21.5–26.2) (p = 0.11) Mean Estimation (Wan's method): 25.2 ± 4.66 vs 23.6 ± 3.78 (p>0.05) |
| Al-Hwary S (2015) [50] | 20 [N.R] vs 40 [N.R] | 46.45 ± 6.29 vs 43.25 ± 5.11 (p>0.05) | N.R | Mean of MELD score: N.R Mean of CP score: N.R CP classification (n,%): All patients are stable cirrhotic patients | Hypertension | Apical 4,2,3-chamber view: -19.98 ± 7.65 vs -29.50 ± 5.92 (p<0.05) | N.R | N.R |
| Sampaio F (2015) [10] | 36 [83.3] vs 8 [62.5] | 54 (48–61) vs 52 (45–54) (p>0.05) | Alcoholic: 21 (58.3) Viral: 10 (27.8) Other causes: 5 (13.9) | Median of MELD score: 9 (7–11) Median of CP Score: 5 (5–7) CP classification (n,%) A: 27 (75) B: 8 (17.8) C: 1 (7.2) | N.R | Apical 4,2,3-chamber view: Median: −18.9 (-16.0 to -20.5) vs -19.0 (-16.1 to -20.6) (p = 0.96) Mean estimation (Wan's method): -18.47 ± 3.47 vs -18.57 ± 4.02 (p = 0.94) | N.R | N.R |
| Anish PG (2019) [51] | 55 [83.63] vs 30 [83.33] | 46.38 vs 45.56 (p>0.05) | N.R | Mean of MELD score: 12 MELD score > 12 (n,%): 22 (40) MELD score < 12 (n,%): 33 (60) Mean of CP score: N.R CP classification (n,%): N.R | Pulmonary artery hypertension: 18 (32.7) Left ventricular hypertrophy: 26 (47.3) | Apical 4,2,3-chamber view: -19.52 ± 2.41 vs -23.66 ± 2.31 (p<0.0001) | N.R | N.R |
| Isaak A (2020) [52] | 42 55 vs 18 [72] | 57 ± 11 vs 54 ± 19 (p>0.05) | Alcoholic: 24 (57) Viral hepatitis: 4 (12) Autoimmune hepatitis: 3 (7) NASH: 3 (7) Hemochromatosis: 1 (2) Congenital anomaly: 1 (2) Cryptogenic: 5 (12) | Mean of MELD score: CP A: 10 ± 2 CP B: 12 ± 5 CP C: 15 ± 5 Mean of CP score: N.R CP classification (n,%) A: 11 (26) B: 20 (48) C: 11 (26) | N.R | Apical 4,2-chamber and parasternal short axis views: -18.5 ± 4.0 ± -22.5 ± 3.6 (p<0.001) | N.R | N.R |

*(Continued)*

**Table 1.** (Continued)

| Author (Year) | Subjects (LC [%male] vs non-LC [% male]) | Baseline Ages (LC vs non-LC, years old) [Median, Q1-Q3 / Mean ± SD]) | Cirrhosis Etiology n (%) | Cirrhosis Prognostic Score and Severity | Cirrhosis Group Comorbidity n (%) | Left Ventricular Longitudinal Strain (LC vs non-LC, %) (Median, Q1-Q3 / Mean ± SD) | Right Ventricular Longitudinal Strain (LC vs non-LC, %) [Median, Q1-Q3 / Mean ± SD] | Other Parameter (TAPSE [mm], RVFAC [%]) (LC vs non-LC) |
|---|---|---|---|---|---|---|---|---|
| **CROSS-SECTIONAL** | | | | | | | | |
| Koç DÖ (2020) [53] | 50 [62] vs 33 [51.5] | 57 ± 13 vs 55 ± 12 (p>0.05) | Viral Hepatitis: 30 (60) NAFLD: 12 (24) Alcoholic: 4 (8) Other cause: 4 (8) | Mean of MELD score: 15.84 ± 7.92 MELD score > 15 (n,%): 25 (50) MELD score < 15 (n,%): 25 (50) Mean of CP score: N.R CP classification (n,%) A: 19 (38) B: 14 (28) C: 17 (34) | N.R | Apical 4,2-chamber view: -19.42 ± 2.83 vs -19.49 ± 2.33 (p>0.05) | Basal, middle, apical segment and ventricular septum view: 17.05 ± 3.49 vs 22.61 ± 0.93 (p = 0.001) | N.R |
| **PROSPECTIVE COHORT** | | | | | | | | |
| Altekin RE (2014) [54] | 38 [63.2] vs 37 [54.1] | 48.3 ± 12.4 vs 45.4 ± 8.6 (p>0.05) | Viral (Hepatitis B and C): 23 (60.5) Cryptogenic: 10 (26.3) Biliary: 5 (13.1) | Mean of MELD score: 11.76 ± 4.92 Mean of CP score: N.R CP classification (n,%): A: 23 (60.5) B: 12 (31.6) C: 3 (7.9) | N.R | Apical 4,2,3-chamber and parasternal short axis view: -20.57 ± 2.1 vs -28.74 ± 3.11 (p<0.001) | N.R | N.R |
| Huang CH (2019) [36] | 80 [80] vs 29 [65.5] | 48.5 (45.0–59.0) vs 49.0 (43.0–52.5) (p>0.05) | Alcoholic: 28 (25.7) Hepatitis B: 22 (20.2) Hepatitis C: 30 (27.5) | Mean of MELD score: Liver cirrhosis with CCM (n = 22): 15.9 ± 8.3 Liver cirrhosis without CCM (n = 57): 15.3 ± 7.9 Mean of CP score: N.R CP classification (n,%): A: 31 (38.75) B/C: 49 (61.25) | Diastolic dysfunction: 27 (34.2) | Apical 4,2,3-chamber view: Median: -21.5 (-22.4 to -20.4) vs -20.2 (-23.0 to -19.1) (p = 0.108) Mean estimation (Wan's method): -21.43 ± 1.51 vs -20.77 ± 3.04 (p = 0.136) | N.R | N.R |
| İnci SD (2019) [55] | 40 [70] vs 26 [61.54] | 46.2 ± 10.1 vs 42.2 ± 8.6 (p>0.05) | N.R | Mean of MELD score: N.R Mean of CP score: N.R CP classification (n,%): C: 40 (100) | N.R | Apical 4-chamber view: -16.0 ± 3.2 vs -17.6 ± 2.2 (p = 0.003) Apical 2-chamber: -16.2 ± 3.3 vs -18.7 ± 2.1 (p = 0.002) | Apical 4- chamber view: -19.2 ± 3.5 vs -21.5 ± 3.6 (p = 0.003) | N.R |
| Özdemir E (2019) [56] | 40 [33] vs 40 [33] | 42.8 ± 8.8 vs 42.5 ± 11.4 (p>0.05) | Hepatitis B = 40 (100) | Mean of MELD score: N.R Mean of CP score: N.R CP classification (n,%): N.R | N.R | Apical 4,2,3-chamber and parasternal short axis view: -19.9 ± 3.4 vs -22.8 ± 1.9 (p<0.001) | N.R | N.R |
| Kim HM (2020) [57] | 33 [75.8] vs 17 [55] | 56.3 ± 9.9 vs 65.0 ± 14.8 (p>0.05) | Viral (Hepatitis B and C): 20 (60.6%) Alcoholic: 9 (27.3) Autoimmune hepatitis: 2 (6.1) Cryptogenic: 2 (6.1) | Mean of MELD score: 18.8 ± 7.4 Mean of CP score: 9.8 ± 2.4 CP classification (n,%) A/B: 10 (30.3) C: 23 (69.7) | Hypertension: 8 (24.2) Diabetes mellitus: 9 (27.3) | Apical 4,2,3-chamber view: -24.2 ± 2.7 vs -18.6 ± 2.2 (p<0.001) | N.R | N.R |

*(Continued)*

**Table 1.** (Continued)

| Author (Year) | Subjects (LC [%male] vs non-LC [% male]) | Baseline Ages (LC vs non-LC, years old) [Median, Q1-Q3 / Mean ± SD]) | Cirrhosis Etiology n (%) | Cirrhosis Prognostic Score and Severity | Cirrhosis Group Comorbidity n (%) | Left Ventricular Longitudinal Strain (LC vs non-LC, %) (Median, Q1-Q3 / Mean ± SD) | Right Ventricular Longitudinal Strain (LC vs non-LC, %) [Median, Q1-Q3 / Mean ± SD] | Other Parameter (TAPSE [mm], RVFAC [%]) (LC vs non-LC) |
|---|---|---|---|---|---|---|---|---|
| **CROSS-SECTIONAL** | | | | | | | | |
| Chen Y (2016) [14] | 103 [74.8] vs 48 [66.7] 103 cirrhotic patients were classified into: Undergoing LTx: 41 Without LTx: 26 Refusing Echo follow-up: 14 Died during study period: 22 | 54.9 ± 7.3 vs 53.5 ± 7.9 (p>0.05) | *Undergoing LTx (n = 41)* Alcoholic: 6 (14.6) Viral: 30 (73.2) Others: 5 (12.2%) *Without LTx (n = 26)* Alcohol: 3 (11.5) Viral: 17 (65.4) Others: 6 (23.1) *Refusing Echo follow-up*: N.R *Died during study period*: N.R | *Undergoing LTx (n = 41)* Mean of MELD score: 21.3 ± 8.9 Mean of CP Score: N.R CP classification (n,%): A: 7 (17.1) B: 11 (26.8) C 23 (56.1) *Without LTx (n = 26)* Mean of MELD score: 12.2 ± 5.6 Mean of CP Score: N.R CP classification (n,%): A: 12 (46.2) B: 10 (38.5) C: 4 (15.3) *Refusing Echo follow-up*: N.R *Died during study period*: N.R | Hypertension: 26 (25.2%) Diabetes mellitus: 21 (20.4) | Apical 4,2,3-chamber view: -18.6 ± 2.6 vs -20.1 ± 2.8 (p<0.01) | Apical 4-chamber view: -21.2 ± 4.4 vs -23.0 ± 2.6 (p<0.01) | TAPSE: 23 ± 4 vs 23 ± 2 (p = 0.77) RVFAC: 53 ± 8 vs 55 ± 6 (p = 0.06) |
| Hassan AA (2019) [58] | 45 [42] vs 30 [53] | 47.13 ± 9.2 vs 46.8 ± 8.9 (p>0.05) | Hepatitis C: 45 (100) | Mean of MELD score: N.R Mean of CP score: N.R CP classification (n,%): A: 15 (33) B: 15 (33) C: 15 (33) | N.R | Apical 4,2,3-chamber view: -19.5 ± 2.7 vs -20.7 ± 4.3 (p = 0.04) | N.R | N.R |
| Ibrahim MG (2020) [59] | 50 [42] vs 50 [38] | 52 ± 12.04 vs 46.76 ± 12.1 (p>0.05) | Hepatitis C: 50 (100) | Mean of MELD score: N.R Mean of CP score: N.R CP classification (n,%): A: 38 (76) B: 12 (24) | Hypertension: 28 (56) Diabetes mellitus: 14 (28) Chronic Hepatitis C infection: 50 (100) | Apical 4,2,3-chamber view: Median: -20 (-26 to -16.5) vs -20 (-28 to -17) Mean estimation (Wan's method): -20.83 ± 7.25 vs -21.67 ± 8.4 (p = 0.59) | Apical 4-chamber view: Median: -22 (-30 to -17) vs -22 (-30 to -17) Mean estimation (Wan's method): -23 ± 9.92 vs -23 ± 9.92 (p = 1.00) | TAPSE: 24.56 ± 3.08 vs 24.06 ± 2.65 (p>0.05) RVFAC: 45.72 ± 4.88 vs 45.64 ± 4.89 (p>0.05) |

ACE-I, angiotensin-converting enzyme inhibitor; ARB, angiotensin receptor blocker; CCB, calcium channel blocker; LC, liver cirrhosis; LTx, liver transplantation; MELD, model for end-stage liver disease; NAFLD, non-alcoholic fatty liver disease; NASH, non-alcoholic steatohepatitis; N.R, not reported; PSC, primary sclerosing cholangitis; RVFAC, right ventricular fractional area change; TAPSE, tricuspid annular plane systolic excursion.

[15], one case-control study [10], and three prospective cohort studies [36,58,59] that reported neutral results. For right ventricular assessment, there were one cross-sectional [47], one case-control [53], and two prospective cohort [14,55] studies showing a significantly lower absolute GLS in patients with cirrhosis than in patients without cirrhosis. Meanwhile, one cross-sectional [15] and one prospective cohort [59] study reported neutral results. Most studies reported an EF > 55% in groups with and without cirrhosis. Considering the standard deviation, six studies [15,45,53,56,58,59] included patients with a borderline EF (50–55%).

## Methodological quality for cross-sectional studies

Methodological quality scores for cross-sectional studies were 8/10 and 9/10 (S3 Table). None of the studies provided information about sample size calculations. The study by Rimbaş et al. [15] excluded 29 of 75 patients with cirrhosis but did not describe the characteristics of these non-respondents. The study by Zhang et al. [47] included only alcoholic cirrhosis; meanwhile, the study by Novo et al. [46] included only patients with hepatitis C etiology and Child-Pugh class A severity cirrhosis. The studies by Zamirian et al. [11]. and von Köckritz et al. [48] included only patients with Child-Pugh class C severity.

## Methodological quality for case-control studies

Methodological quality scores for the case-control studies were 7/9 and 8/9 (S4 Table). None of these case-control studies provided a clear description of the non-response rate. Sampaio et al. [49] and Isaak et al. [52] reported the number of excluded participants in the group with cirrhosis, but not in the control group, and a large discrepancy in the population was evident between the case and control group. In addition to a lack of reports concerning the non-response rate, Al-Hwary et al. [50] did not provide any information regarding the validation of the cirrhosis diagnosis.

## Methodological quality for cohort studies

Methodological quality scores ranged from 6/9 to 9/9 (S5 Table). The study by Altekin et al. [54] scored 6/9 because it excluded alcoholic cirrhosis and thus did not cover the entire population of cirrhotic liver disease. In addition, this study had no description of the non-exposed cohort derivation and a shorter follow-up period (less than 6 months) compared with other studies (at least 1 year). Meanwhile, the study by Kim et al. [57] scored 6/9 because there was no description of the source of the non-exposed cohort and no adjustment for confounders, stratifications, or matching to improve comparability between the group with cirrhosis and control. The study by İnci et al. [55] only included patients with Child–Pugh C class severity cirrhosis and did not report the length of follow-up. Özdemir et al. [56] only included patients with hepatitis B viral infection etiology and did not report the source of non-exposed group derivation. Chen et al. [14] reported a different source of population derivation between the exposed and non-exposed cohorts. Hassan et al. [58] evaluated only the group with hepatitis C viral infection cirrhosis etiology and did not report the source of non-exposed group derivation. The study by Ibrahim et al. [59] included only patients with hepatitis C viral etiology cirrhosis.

## Left ventricular global longitudinal strain in patients with cirrhosis versus patients without cirrhosis

We evaluated 19 studies [10,11,14,15,36,45,46,48–53,55–59] reporting LV-GLS from patients with and without cirrhosis. The pooled analysis of LV-GLS, evaluated using 2D-STE in17 studies and CMR in two studies, revealed a significantly lower absolute LV-GLS based on the mean difference in patients with cirrhosis (MD:-1.43; 95%CI,-2.79 to -0.07; p = 0.04) (Fig 2). REM was considered to be due to the significant statistical heterogeneity between studies ($I^2$ = 95%, p<0.00001).

We conducted sensitivity analysis to evaluate the stability of our findings. The pooled results changed several times after each omission of one study from the meta-analysis (S6 Table). The omission of 10 studies [14,45,46,49–52,54–56] individually revealed insignificance in the pooled meta-analysis. Meanwhile, omission of the other nine studies

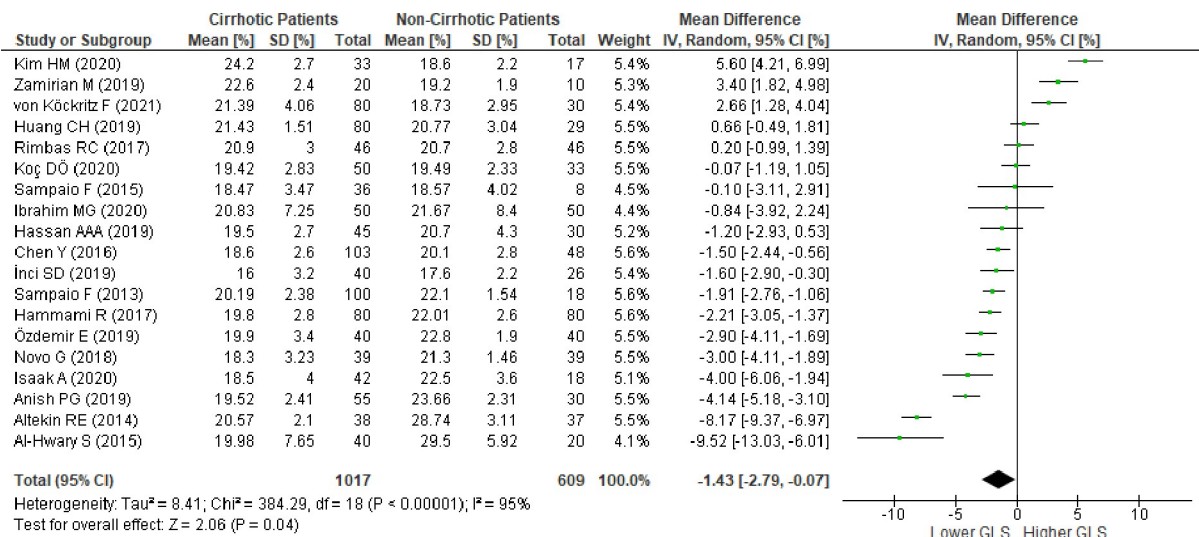

**Fig 2. Mean difference of left ventricular global longitudinal strain in patiens with cirrhosis versus patients without cirrhosis evaluated using the random effect model.** SD, standard deviation; IV, inverse variance; CI, confidence interval; df, degrees of freedom; Chi$^2$, chi-squared statistic; p, p-value; I$^2$, I-squared heterogeneity statistic; Z, Z statistic.

[10,11,15,36,48,53,57–59] resulted in a significantly lower absolute LV-GLS in favor of the group with cirrhosis. We found that the omission of the study by Altekin et al. [54] resulted in the largest Q and Tau$^2$ change. The results showed no significant absolute LV-GLS difference (MD:-1.02; 95%CI,-2.21 to 0.18; p = 0.09; I$^2$ = 93%, p<0.00001) (S1 Fig). Further sensitivity analysis was performed, and the omission of the study by Kim et al. [57] showed the largest change in Q and Tau$^2$ (S7 Table). Despite the persistent heterogeneity, the final result successfully provided stability in the next sensitivity analysis (S8 Table), resulting in a significantly lower absolute LV-GLS favoring the group with cirrhosis (MD:-1.39; 95%CI,-2.40 to -0.37; p = 0.008; I$^2$ = 90%, p<0.00001) (S2 Fig). To achieve a heterogeneity of less than 75%, seven studies [11,36,48,50,51,54,57] were omitted, and the pooled meta-analysis from the resulting 12 studies showed a significantly lower absolute LV-GLS favoring the group with cirrhosis (MD:-1.66; 95%CI,-2.33 to -1.00; p <0.00001; I$^2$ = 69%, p = 0.0002) (Fig 4). Moreover, to evaluate whether cardiovascular comorbidities could contribute to a decreased LV-GLS, we excluded studies that included patients with hypertension, diabetes mellitus, and left ventricular hypertrophy. Further exclusion of three studies [14,46,59] that included patients with hypertension and diabetes mellitus showed similar results (MD:-1.55; 95%CI,-2.37 to -0.73, p = 0.0002; I$^2$ = 73%, p = 0.0003), while exclusion of the study by Hammami et al. [45] that included patients with left ventricular hypertrophy showed simillar results (MD:-1.59; 95%CI,-2.34 to -0.84, p<0.00001; I$^2$ = 70%, p = 0.0002). Other comorbidities, such as coronary artery disease and valvular heart disease, were excluded in all studies; therefore, these variables did not influence the current results. Additionally, the exclusion of two studies [15,49] which included patients with diastolic dysfunction revealed a similar result, indicating a significant reduction in LV-GLS in patients with cirrhosis (MD:-1.85; 95%CI,-2.57 to -1.33, p<0.00001; I$^2$ = 63%, p = 0.003). In order to evaluate whether ongoing medical therapy (diuretics, beta-blockers, angiotensin-converting enzyme inhibitor [ACE]-I/angiotensin receptor blocker [ARB], calcium channel blocker [CCB]) could influence LV-GLS, we excluded three studies [10,14,45] that reported these medications in patients with cirrhosis, and the result showed that there was still a significant reduction of LV-GLS in patients with cirrhosis (MD:-1.69; 95%

CI,-2.59 to -0.79, p = 0.0002; $I^2$ = 76%, p<0.0001). The effect of diuretic treatment seen in the study by Sampaio et al. [10] was excluded from the analysis, resulting in a similar result (MD:-1.72; 95%CI,-2.40 to -1.04, p<0.00001; $I^2$ = 71%, p = 0.0002). Excluding the study by Hammami et al. [45] to evaluate the influence of beta-blocker treatment on the results also showed similar findings (MD:-1.59; 95%CI,-2.34 to -0.84, p<0.0001; $I^2$ = 70%, p = 0.0002).

## Right ventricular global longitudinal strain in patients with cirrhosis versus patients without cirrhosis

We evaluated six studies [14,15,47,53,55,59] reporting RV-GLS in patients with and without cirrhosis. The pooled analysis of RV-GLS, evaluated using 2D-STE in all studies, revealed a significantly lower absolute RV-GLS based on the mean difference in patients with cirrhosis (MD:-1.95; 95%CI,-3.86 to -0.05; p = 0.04) (Fig 3). REM was considered because there was significant statistical heterogeneity between the studies ($I^2$ = 90%,p<0.00001).

We conducted sensitivity analysis to evaluate the stability of our findings. When omitting each study individually from the meta-analysis, the pooled results changed several times (S9 Table). The successive omission of three studies [14,47,55] revealed insignificance in the pooled meta-analysis. Meanwhile, omission of the other three studies [15,53,59] resulted in a significantly lower absolute RV-GLS in favor of the group with cirrhosis. We found that omission of the study by Koç et al. [53] resulted in the largest Q and $Tau^2$ value change. The omission resulted in a lower absolute RV-GLS, favoring the group with cirrhosis (MD:-1.30; 95% CI,-2.43 to -0.18; p = 0.02; $I^2$ = 57%, p = 0.05) (S3 Fig). Further sensitivity analysis was performed, and the study by Rimbaş et al. [15] showed the largest change in Q and $Tau^2$ values (S10 Table). The final result showing no heterogeneity successfuly gives stability in the next sensitivity analysis as shown in S11 Table and resulted in a significantly lower absolute RV-GLS favoring the group with cirrhosisp (MD:-1.90; 95%CI,-2.62 to -1.18; p<0.0001; $I^2$ = 0%, p = 0.76) (S4 Fig). Moreover, to evaluate whether cardiovascular comorbidities could contribute to a decreased RV-GLS, we excluded studies that included patients with hypertension, diabetes mellitus, and left ventricular hypertrophy. Further exclusion of two studies [14,59] with patients with hypertension and diabetes mellitus showed similar results (MD:-2.09; 95% CI,-3.05 to -1.13, p<0.0001; $I^2$ = 0%, p = 0.78). Among these studies, no study had evaluated RV-GLS in patients with left ventricular hypertrophy. Other comorbidities, such as coronary artery disease and valvular heart disease, were excluded in all studies; therefore, these variables did not influence the current results. The effect of diastolic dysfunction on RV-GLS results could not be evaluated because of the limited data available in the remaining study. The effect of ongoing medical therapy (diuretics, beta-blockers, ACE-I/ARB, CCB) on RV-GLS results was evaluated by excluding a study by Chen et al. [14], which still showed a significant

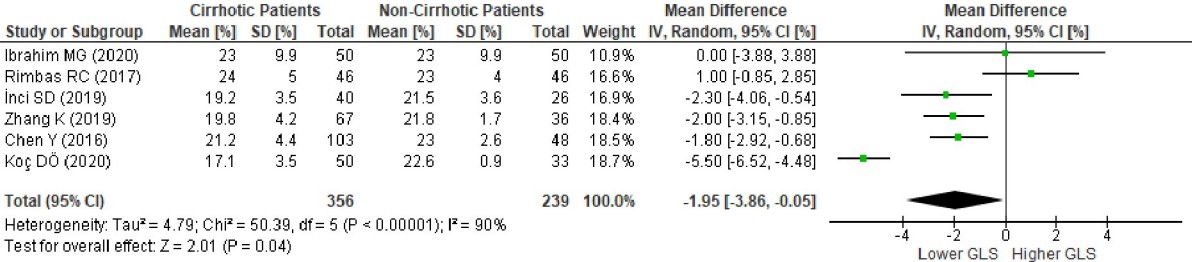

**Fig 3. Mean difference of right ventricular global longitudinal strain in patientw with cirrhosis versus patients without cirrhosis evaluated using the random effect model.** SD, standard deviation; IV, inverse variance; CI, confidence interval; df, degrees of freedom; $Chi^2$, chi-squared statistic; p, p-value; $I^2$, I-squared heterogeneity statistic; Z, Z statistic.

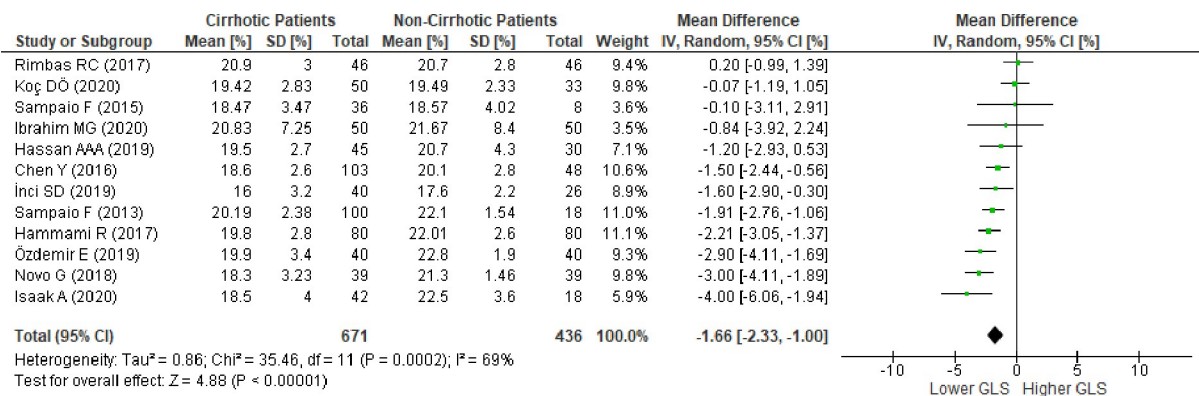

**Fig 4. Mean difference of left ventricular global longitudinal strain in patients with cirrhosis versus patients without cirrhosis after consecutive omission of studies by Altekin, et al.; Kim, et al; Zamirian, et al; von Köckritz, et al.; Anish, et al.; Al-Hwary, et al.; and Huang, et al.** SD, standard deviation; IV, inverse variance; CI, confidence interval; df, degrees of freedom; Chi$^2$, chi-squared statistic; p, p value; I$^2$, I-squared heterogeneity statistic; Z, Z statistic.

reduction in RV-GLS in patients with cirrhosis (MD:-1.97; 95%CI,-2.90 to -1.04, p<0.0001; I$^2$ = 0%, p = 0.57).

## Subgroup analysis

Subgroup analysis based on study design showed that when studies were grouped into cross-sectional, case-control, or cohort designs, no significant difference in LV-GLS was found between groups of cross-sectional and cohort studies (S5 Fig). The pooled analysis from all cross-sectional studies did not show a significant difference of LV-GLS between groups (MD:0.16; 95%CI,-2.18–2.50; p = 0.89; I$^2$ = 95%). Pooled analysis of all prospective cohort studies showed similar results (MD:-1.26; 95%CI -3.97–1.46; p = 0.36; I$^2$ = 97%). Meanwhile, a significantly lower LV-GLS in the group with cirrhosis was observed in the pooled analysis of all case-control studies (MD:-3.00; 95%CI,-4.88 to -1.12; p = 0.002; I$^2$ = 90%).

Sensitivity analysis conducted on each study design showed that LV-GLS was significantly lower in the group with cirrhosis (MD:-1.78; 95%CI,-2.50 to -1.07; p<0.0001; I$^2$ = 70%, p = 0.0003) (Fig 5). No significant differences were detected between subgroups (p = 0.18,I$^2$ = 42.5%). Subgroup analysis by study design for RV-GLS was not performed because of the limited number of studies available.

We did not perform an analysis based on age because all included studies presented age as aggregate information that was reported using mean or median; thus, it could introduce an ecological bias [33]. Subgroup analysis based on sex was not able to be performed because no study reported GLS results separately between male and female patients with cirrhosis.

We did not perform a meta-analysis based on the individual Child-Pugh classification or MELD scores due to the limited number of studies [45,51–53]. Subgroup analysis by dividing cirrhotic patients into compensated and decompensated groups was not possible due to limited studies. There were only two studies [46,50] in which all included patients were classified as having compensated cirrhosis,and three studies [11,48,55] in which all patients included were classified as having decompensated cirrhosis. Comparing GLS changes between studies that included a majority (> 50%) of compensated patients [10,15,46,50,54,59] with studies that included a majority (>50%) of decompensated patients [11,14,36,45,48,49,52,53,55,57,58] may also introduce ecological bias.

Subgroup analysis based on etiology was not performed because of the limited number of studies [15,36,48,53]. There was only one study [47] in which the patient inclusion criteria

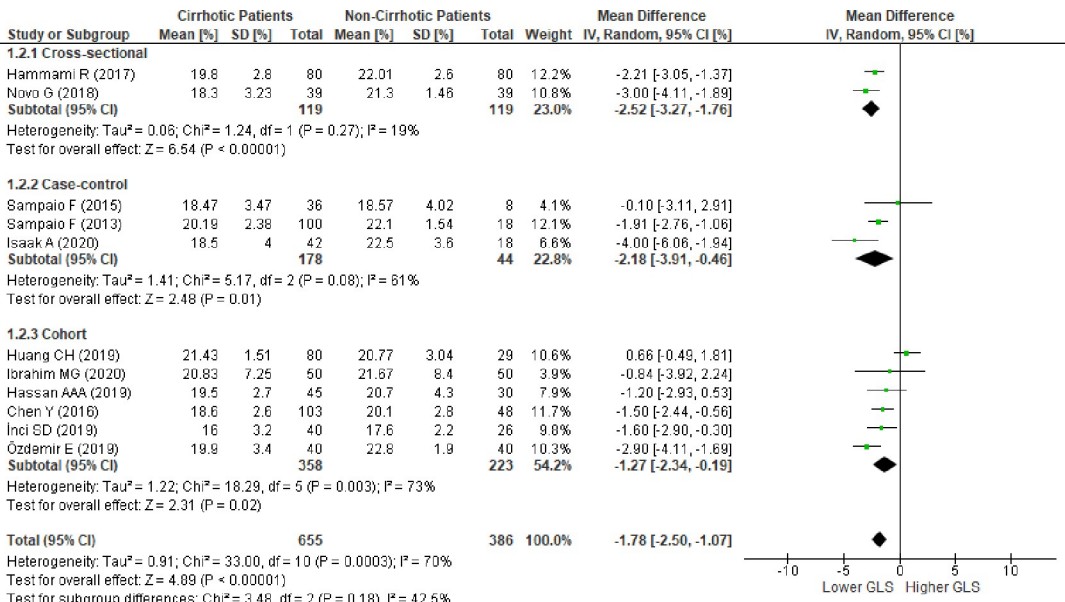

**Fig 5. Mean difference of left ventricular global longitudinal strain in patients with cirrhosis versus patients without cirrhosis after subgroup analysis according to study design and consecutive omission of studies by Zamirian, et al.; von Köckritz, et al. and Rimbaş, et al. (cross sectional); consecutive omission of studies by Al-Hwary, et al.; Koç, et al.; and Anish, et al. (case control); and consecutive omission of studies by Altekin, et al.; and Kim, et al (cohort).** SD, standard deviation; IV, inverse variance; CI, confidence interval; df, degrees of freedom; Chi2, chi-squared statistic; p, p value; $I^2$, I-squared heterogeneity statistic; Z, Z statistic.

were solely due to alcoholic etiology, one study [56] was solely due to hepatitis B viral infection, and three studies [46,58,59] solely due to hepatitis C viral infection. Based on individual results from our included studies comparing GLS between various etiologies, two studies reported no significant differences in absolute LV-GLS [48,53] or RV-GLS [53] between different cirrhosis etiologies. However a study by Huang et al. [36] showed that alcoholic etiology had a significantly lower absolute LV-GLS than etiology due to hepatitis B and C infection (20.6±2.3 vs 22.4±2.5 vs 21.9±1.6, respectively; p = 0.034).

## Other parameters

We found no significant difference in tricuspid annular plane systolic excursion (TAPSE) between patients with cirrhosis and patients without cirrhosis based on all available data, which was reported by six studies (MD:0.30; 95%CI,-0.25–0.85, p = 0.29; $I^2$ = 0%,p = 0.59) (S6 Fig) [14,15,46,47,49,59]. Similarly, all four studies [14,15,47,59] reported that right ventricular fractional area change (RVFAC) between patients with and without cirrhosis showed no significant difference (MD:-0.56; 95%CI,-1.85–0.73, p = 0.40; $I^2$ = 0%, p = 0.44) (S7 Fig).

## Publication bias

A funnel plot (S8 Fig) was generated from the subgroup analysis of studies evaluating LV-GLS. Egger's regression test revealed that the Z value was 0.490 (p = 0.624), while the rank correlation test indicated a Kendall's tau of 0.164 (p = 0.542). The results of the trim-and-fill analysis are shown in S9 Fig. The adjusted pooled mean difference with an addition of two studies to make the funnel plot balance was -2.16 (95%CI:-3.00 to -1.32). Overall, a slight publication bias may exist.

### Fail-safe N

Based on all the meta-analyses performed, Rosenthal's fail-safe number showed robust results. Meta-analysis of LV-GLS (Fig 2) and RV-GLS (Fig 3) from all studies showed a fail-safe number of 575 and 123 unpublished studies with null findings, respectively, until a non-significant effect size was obtained. Meta-analysis of 12 studies to reach I2 less than 75% (Fig 4) showed a fail-safe number of 336. The sensitivity analysis conducted on each study design (Fig 5) resulted in a fail-safe number of 317.

### Meta-regression

Meta-regression analysis performed to explain variations in the reduction of LV-GLS in the group with cirrhosis revealed that the MELD score and proportion of decompensated cirrhosis covariates were significant and explained $R^2 = 28\%$ and $R^2 = 19\%$ of the heterogeneity in LV-GLS reduction, respectively. Figs 6 and 7 show the covariate effect graphs and $R^2$ calculations, respectively. Meta-regression analysis showed that study design, proportion of male subjects, age, proportion of alcoholic etiology, proportion of viral etiology, baseline LVEF, and NOS score had no influence on LV-GLS reduction in the goup with cirrhosis. All data results of the meta-regression are shown in S2–S20 Tables.

## Discussion

The group of patients with cirrhosis had a 1.66% lower LV-GLS after sensitivity analysis was performed by excluding seven studies [11,36,48,50,51,54,57]. Studies by Altekin et al. [54] and Kim et al. [57] contributed the most to the heterogeneity, which may be related to the low methodological quality of the studies. Furthermore, Kim et al. [57] showed a significant age difference between groups; thus, differences in methodology may have affected the outcomes. Meanwhile, a 1.90% lower RV-GLS was found in the group of patients with cirrhosis after two studies were omitted to reach 0% heterogeneity. Heterogeneity sources are from Koç et al. [53] and Rimbaş et al. [15], which might be due to the different study designs and methodological quality.

We realized that the studies included in our analysis had different age ranges, sex proportions, etiology and severity of cirrhosis, and study designs. These variables may have affected the outcomes of our meta-analysis. At present, only subgroup analysis based on study design could be conducted due to the limited number of studies available for subgroup analysis. According to Harrer et al. [33], at least 10 studies should be available for a powerful analysis to show that the subgroups are equivalent, due to the dependence of subgroup analysis on statistical power. After conducting a sensitivity analysis by omitting several studies with low methodological quality, which contributed the most to heterogeneity, the meta-analysis showed that pooled results based on each study design were consistent with the lower LV-GLS in the group of patients with cirrhosis. We reduced heterogeneity by 25% while preserving a meaningful outcome, with a lower absolute LV-GLS of 1.78% in the group with cirrhosis. This indicates that methodological quality and heterogeneity may have affected the results.

We performed meta-regression to investigate the true cause of heterogeneity influencing high $I^2$ values. The MELD score and proportion of decompensated cirrhosis were significantly related to GLS reduction in the group with cirrhosis. These factors may have contributed to the high heterogeneity of the meta-analysis results. We found no significant influence on the results from the study design, sex, age, etiology, baseline LVEF, or NOS score. The finding that cirrhosis severity influences GLS is supported by a previous study [39]. While a lower GLS indicated subclinical systolic dysfunction, a higher GLS was linked to more severe liver disease [39]. Additionally, although some of our meta-analyses showed borderline significant results,

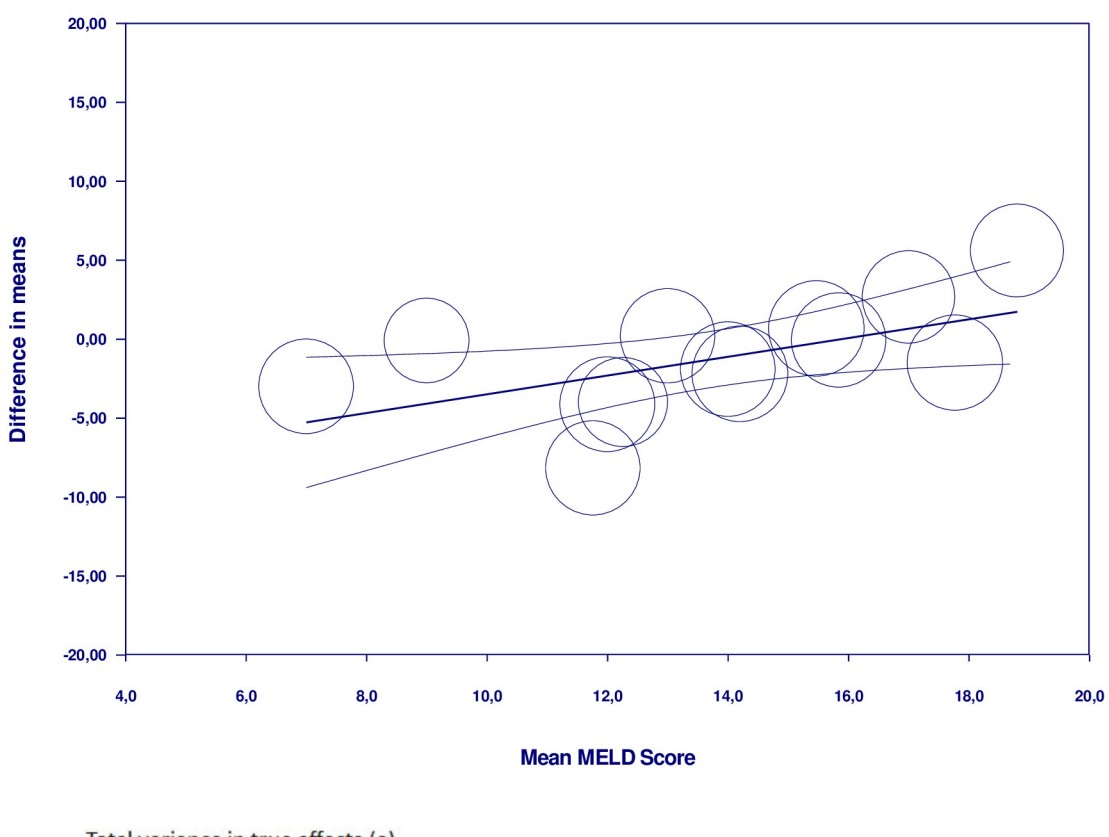

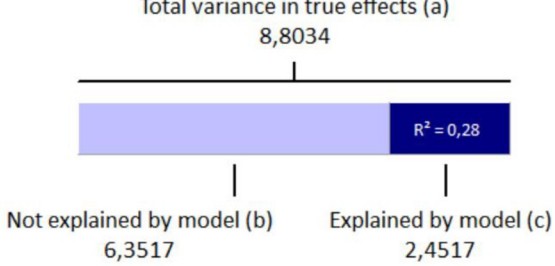

(a) To compute the total variance (of all studies about the grand mean) we run the regression with no covariates.
(b) To compute the variance not explained by the model (of all studies about the regression line) we run the regression with the covariates.
(c) The difference between these values gives us the variance explained by the model.

**Fig 6. Meta regression result for MELD score covariate significantly influencing GLS reduction in cirrhotic group with R² = 28%.**

our fail-safe number analysis confirmed that all results are robust. This is because the number of studies that are needed to change the result to insignificant exceeded the minimum limit, as proposed by Rosenthal et al [30].

Our study has some limitations. First, due to the limited number of studies, we were unable to provide a subgroup analysis based on age, sex, cirrhosis etiology, and severity. Second, our meta-analysis, which pooled directly from 19 studies for LV-GLS and six studies for RV-GLS, showed considerable statistical heterogeneity. The source of heterogeneity was more likely due to the differences found between studies and not merely due to sampling errors. Major clinical

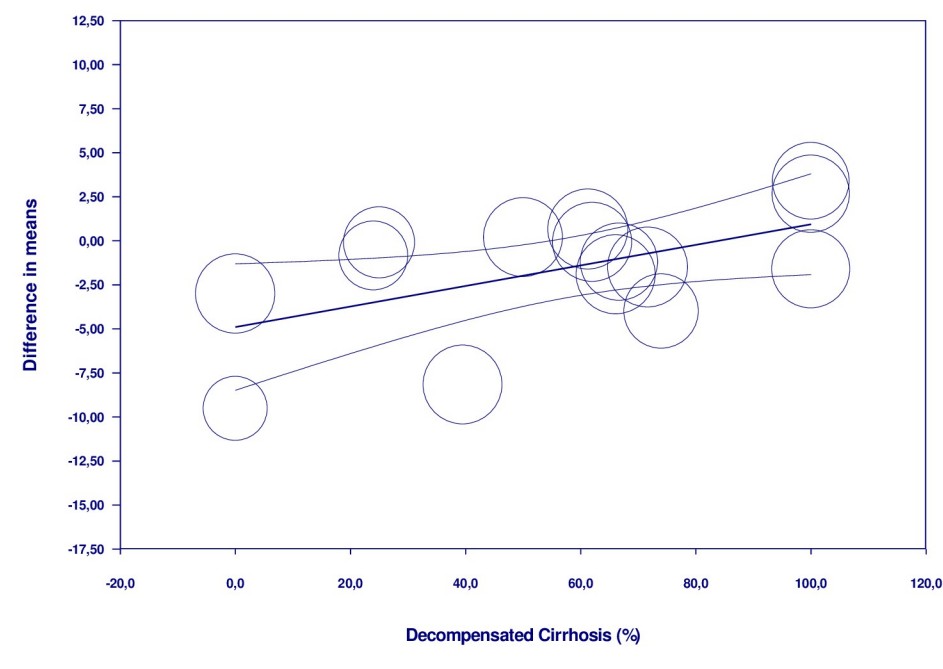

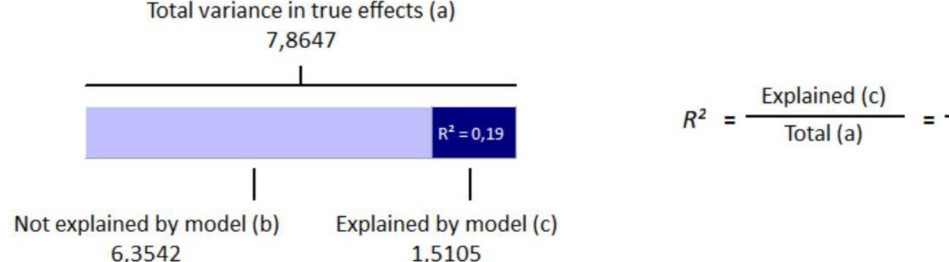

(a) To compute the total variance (of all studies about the grand mean) we run the regression with no covariates.
(b) To compute the variance not explained by the model (of all studies about the regression line) we run the regression with the covariates.
(c) The difference between these values gives us the variance explained by the model.

**Fig 7. Meta regression result for decompensated cirrhosis proportion covariate significantly influencing GLS reduction in cirrhotic group with $R^2$ = 19%.**

heterogeneity was due to cirrhosis severity based on the meta-regression. Thus, further studies should explore GLS according to the cirrhosis severity.

## Conclusion

Our meta-analysis showed that patients with cirrhosis had a lower GLS than controls. However, cirrhosis severity was significantly related to GLS reduction. Future studies should provide further analysis to elucidate a significant difference in GLS in cirrhotic patients according to cirrhosis severity.

## Supporting information

**S1 Checklist.**
(DOCX)

**S1 Fig. The mean difference in left ventricular global longitudinal strain from patients with and without cirrhosis after omission of the study by Altekin et al.** SD, standard deviation; IV, inverse variance; CI, confidence interval; df, degrees of freedom; $Chi^2$, chi-squared statistic; p, p-value; $I^2$, I-squared heterogeneity statistic; Z, Z statistic.
(DOCX)

**S2 Fig. The mean difference in left ventricular global longitudinal strain between patients with and withour cirrhosis after consecutive omission of studies by Altekin et al. and Kim et al.** SD, standard deviation; IV, inverse variance; CI, confidence interval; df, degrees of freedom; $Chi^2$, chi-squared statistic; p, p-value; $I^2$, I-squared heterogeneity statistic; Z, Z statistic.
(DOCX)

**S3 Fig. The mean difference in right ventricular global longitudinal strain from patients with and withour cirrhosis after omission of study by Koç DÖ, et al.** SD, standard deviation; IV, inverse variance; CI, confidence interval; df, degrees of freedom; $Chi^2$, chi-squared statistic; p, p-value; $I^2$, I-squared heterogeneity statistic; Z, Z statistic.
(DOCX)

**S4 Fig. The mean difference in right ventricular global longitudinal strain from patients with and without cirrhosis after consecutive omission of studies by Koç DÖ, et al and Rimbaş, et al.** SD, standard deviation; IV, inverse variance; CI, confidence interval; df, degrees of freedom; $Chi^2$, chi-squared statistic; p, p-value; $I^2$, I-squared heterogeneity statistic; Z, Z statistic.
(DOCX)

**S5 Fig. The mean difference in left ventricular global longitudinal strain from patients with and withour cirrhosis after subgroup analysis according to study design.** SD, standard deviation; IV, inverse variance; CI, confidence interval; df, degrees of freedom; $Chi^2$, chi-squared statistic; p, p-value; $I^2$, I-squared heterogeneity statistic; Z, Z statistic.
(DOCX)

**S6 Fig. The mean difference in tricuspid annular plane systolic excursion between patients with and without cirrhosis.** SD, standard deviation; IV, inverse variance; CI, confidence interval; df, degrees of freedom; $Chi^2$, chi-squared statistic; p, p-value; $I^2$, I-squared heterogeneity statistic; Z, Z statistic.
(DOCX)

**S7 Fig. The mean difference in right ventricular fractional area change between patients with and withour cirrhosis.** SD, standard deviation; IV, inverse variance; CI, confidence interval; df, degrees of freedom; $Chi^2$, chi-squared statistic; p, p-value; $I^2$, I-squared heterogeneity statistic; Z, Z statistic.
(DOCX)

**S8 Fig. Funnel plot for publication bias on studies evaluating left ventricular global longitudinal strain based on subgroup analysis according to study design in Fig 5.**
(DOCX)

**S9 Fig. Trim-and-fill analysis for the funnel plot in Fig 5.** GEN generated the mean difference.
(DOCX)

**S10 Fig. Meta-regression result for study design covariate have no influence on GLS reduction in group with cirrhosis with $R^2 = 0\%$.**
(DOCX)

**S11 Fig. Meta-regression result for proportion of male subjects covariate have no influence on GLS Reduction in group with cirrhosis with $R^2 = 0\%$.**
(DOCX)

**S12 Fig. Meta-regression result for age covariate have no influence on GLS reduction in group with cirrhosis with $R^2 = 3\%$.**
(DOCX)

**S13 Fig. Meta-regression result for proportion of alcoholic etiology covariate have no influence on GLS reduction in group with cirrhosis with $R^2 = 0\%$.**
(DOCX)

**S14 Fig. Meta-regression result for proportion of viral etiology covariate have no influence on GLS reduction in group with cirrhosis with $R^2 = 0\%$.**
(DOCX)

**S15 Fig. Meta-regression result for baseline left ventricular ejection fraction in group with cirrhosis have no influence on GLS reduction in group with cirrhosis with $R^2 = 0\%$.**
(DOCX)

**S16 Fig. Meta-regression result for Newcastle Ottawa Scale Score have no influence on GLS reduction in group with cirrhosis with $R^2 = 0\%$.**
(DOCX)

**S1 Table. Search strategy completed using MeSH terms and [all fields].**
(DOCX)

**S2 Table. Complete characteristic of included study with left and right ventricular global longitudinal strain results.**
(DOCX)

**S3 Table. Newcastle Ottawa Scale for cross-sectional studies.**
(DOCX)

**S4 Table. Newcastle Ottawa Scale for case-control studies.**
(DOCX)

**S5 Table. Newcastle Ottawa Scale for cohort studies.**
(DOCX)

**S6 Table. Sensitivity analysis for mean difference of left ventricular global longitudinal strain from cirrhotic versus non-cirrhotic patients.**
(DOCX)

**S7 Table. Sensitivity analysis for mean difference of left ventricular global longitudinal strain from cirrhotic versus non-cirrhotic patients after omission of the study by Altekin et al.**
(DOCX)

**S8 Table. Sensitivity analysis for mean difference of left ventricular global longitudinal strain from cirrhotic versus non-cirrhotic patients after omission of studies by Altekin**

**et al. and Kim et al.**
(DOCX)

**S9 Table. Sensitivity analysis for mean difference of right ventricular global longitudinal strain from cirrhotic versus non-cirrhotic patients.**
(DOCX)

**S10 Table. Sensitivity analysis for mean difference of right ventricular global longitudinal strain from cirrhotic versus non-cirrhotic patients after the mission of study by Koç et al.**
(DOCX)

**S11 Table. Sensitivity analysis for mean difference of right ventricular global longitudinal strain from cirrhotic versus non-cirrhotic patients after mission of study by Koç et al.. and Rimbaş et al.**
(DOCX)

**S12 Table. Meta-regression results and $R^2$ for study design covariate.**
(DOCX)

**S13 Table. Meta-regression results and $R^2$ for proportion of male subjects covariate.**
(DOCX)

**S14 Table. Meta-regression results and $R^2$ for age covariate.**
(DOCX)

**S15 Table. Meta-regression results and $R^2$ for MELD score covariate.**
(DOCX)

**S16 Table. Meta-regression results and $R^2$ for proportion of decompensated cirrhosis covariate.**
(DOCX)

**S17 Table. Meta-regression results and $R^2$ for proportion of alcholic etiology covariate.**
(DOCX)

**S18 Table. Meta-regression results and $R^2$ for proportion of viral etiology covariate.**
(DOCX)

**S19 Table. Meta-regression results and $R^2$ for baseline left ventricular ejection fraction in cirrhotic group covariate.**
(DOCX)

**S20 Table. Meta-regression results and R2 for Newcastle Ottawa Scale score covariate.**
(DOCX)

**S1 Raw data.**
(XLSX)

## Acknowledgments

### Ethics approval

No studies with human participants or animals were performed by any of the authors. All the studies were performed in accordance with the ethical standards of each study.

## Permission to reproduce material from other sources

The authors warrant that this manuscript contains no material that infringes on the copyright of any other person(s)/source(s).

## Author Contributions

**Conceptualization:** Denio A. Ridjab, Ignatius Ivan, Fanny Budiman, Riki Tenggara.

**Data curation:** Denio A. Ridjab, Ignatius Ivan, Fanny Budiman.

**Formal analysis:** Denio A. Ridjab, Ignatius Ivan, Fanny Budiman.

**Funding acquisition:** Denio A. Ridjab, Riki Tenggara.

**Investigation:** Denio A. Ridjab, Ignatius Ivan, Fanny Budiman, Riki Tenggara.

**Methodology:** Denio A. Ridjab, Ignatius Ivan, Fanny Budiman.

**Project administration:** Denio A. Ridjab, Ignatius Ivan.

**Resources:** Denio A. Ridjab, Ignatius Ivan, Fanny Budiman.

**Software:** Denio A. Ridjab, Ignatius Ivan.

**Supervision:** Denio A. Ridjab, Riki Tenggara.

**Validation:** Denio A. Ridjab, Riki Tenggara.

**Visualization:** Denio A. Ridjab, Ignatius Ivan, Fanny Budiman.

**Writing – original draft:** Denio A. Ridjab, Ignatius Ivan, Fanny Budiman.

**Writing – review & editing:** Denio A. Ridjab, Ignatius Ivan, Fanny Budiman, Riki Tenggara.

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
