## [Decision Letter · Decision Letter 0]

8 Feb 2022

PONE-D-21-37272Evaluation of Subclinical Ventricular Systolic Dysfunction Assessed Using Global Longitudinal Strain in Liver Cirrhosis: A Systematic Review and Meta-Analysis with Trial Sequential AnalysisPLOS ONE

Dear Dr. Ridjab,

Thank you for submitting your manuscript to PLOS ONE. After careful consideration, we feel that it has merit but does not fully meet PLOS ONE’s publication criteria as it currently stands. Therefore, we invite you to submit a revised version of the manuscript that addresses the points raised during the review process.

ACADEMIC EDITOR: All issues raised by expert reviewers are required. Moreover, the authors should revise statistical approach in order to better support the conclusions.

We look forward to receiving your revised manuscript.

Kind regards,

Vincenzo Lionetti, M.D., PhD

Academic Editor

PLOS ONE

Journal Requirements:

Reviewers' comments:

Reviewer's Responses to Questions

**Comments to the Author**

1. Is the manuscript technically sound, and do the data support the conclusions?

Reviewer #1: Yes

Reviewer #2: Yes

2. Has the statistical analysis been performed appropriately and rigorously? 

Reviewer #1: I Don't Know

Reviewer #2: I Don't Know

3. Have the authors made all data underlying the findings in their manuscript fully available?

Reviewer #1: Yes

Reviewer #2: Yes

4. Is the manuscript presented in an intelligible fashion and written in standard English?

Reviewer #1: Yes

Reviewer #2: Yes

5. Review Comments to the Author

Reviewer #1: Dear Authors,

Thank you for the opportunity to read your work.

After read it with attention the results seems to me sound.

I'm not totally able to assess the statistical analysis, but it appear to me to be right.

Please spent time to siimplify the text that is now too technical.

I will read your revision version with pleasure.

Best Regards

Reviewer #2: The Authors performed a detailed metanalysis showing a significant reduction of LV-GLS and RV-GLS in cirrhotic patients. Overall, they performed an extensive search and analysis from as many as 5347 studies, yielding 20 studies (1738 pts; 2 studies with CMR, 18 with echo). Please find some comments:

-All patients presented by definition a LVEF >50%, but it is unknown how many patients presented cardiovascular comorbidities (hypertension ,diabetes, valvular heart disease, coronary artery disease, LV hypertrophy....) that might contribute to a decreased LV or RV GLS

-It is not clear which was the relationship between LVGLS and LV ejection fraction/MAPSE, or between RVGLS and TAPSE/FAC; did you analyse this point? Was there a difference between patients with bordeline LVEF and normal/high LVEF?

-Besides an analysis on compensated/decompensated liver cirrhosis, were there differences according to ongoing medical therapy (diuretics? beta-blockers? ACE/ARB? MRA?) or to the presence of heart failure?

6. PLOS authors have the option to publish the peer review history of their article (what does this mean?). If published, this will include your full peer review and any attached files.

Reviewer #1: **Yes: **Luigi Vetrugno

Reviewer #2: No

---

## [Author Response · Author response to Decision Letter 0]

16 Mar 2022

Response from Academic Editor:

ACADEMIC EDITOR: All issues raised by expert reviewers are required. Moreover, the authors should revise statistical approach in order to better support the conclusions.

Response from Author:

We have revise our statistical approach. The reporting of the trial sequential analysis has been improved by defining the mean difference a priori, before the trial sequential analysis run while also correcting the heterogeneity to reach a low heterogeneity (at least 25%). With this approach we report a more robust result. This was further described in the results section (Trial Sequential Analysis) and indicated in Figure S10 and Figure S11. This means that all result from meta analysis have shown a conclusive evidence with a low risk of a false positive result. 

We calculated the information size required based on α = 5% significance level, and β = 20% (80% power) with an anticipated LV-GLS change of -1.43% and RV-GLS change of -1.95% in favor of cirrhotic groups. Heterogeneity was corrected to reach I2 of at least 25% (low heterogeneity). [PAGE 4]

TSA result for LV-GLS and RV-GLS are shown in S10 Fig and S11 Fig, respectively. Both TSA showed that these studies cross trial monitoring boundary and confirm a lower absolute GLS in favor of cirrhotic group. This indicate that result of GLS in both ventricle have low risk of false positive results. After conducting TSA with sensitivity analysis that has been done previously in Fig 4 for LV-GLS and Figure S4 for RV-GLS, a further TSA is then generated for LV-GLS (S12 Fig) and RV-GLS (S13 Fig). From these TSA, the result is also conclusive and have low risk of false positive results. [PAGE 12]

The conclusion have been better improved in accordance with the improved results by using a revised statistical approach

Despite the clinical and methodological heterogeneity, TSA revealed that current meta analysis have shown firm evidences that cirrhotic patients have a lower GLS when compared with controls. Future studies should provide a further analysis to elucidate whether there will be a significant difference of GLS in cirrhotic patients with different age, sex, cirrhosis severity and etiologies. [PAGE 16]

Response from Reviewer:

Reviewer #1: Dear Authors, Thank you for the opportunity to read your work. After read it with attention the results seems to me sound. I'm not totally able to assess the statistical analysis, but it appear to me to be right. Please spent time to siimplify the text that is now too technical. I will read your revision version with pleasure.

Response from Authors:

We have simplify the text particularly the discussion part in order to be more readable. Several grammatical errors have also been corrected. [PAGE 2-16]

Response from Reviewer:

Reviewer #2: The Authors performed a detailed metanalysis showing a significant reduction of LV-GLS and RV-GLS in cirrhotic patients. Overall, they performed an extensive search and analysis from as many as 5347 studies, yielding 20 studies (1738 pts; 2 studies with CMR, 18 with echo). Please find some comments:

-All patients presented by definition a LVEF >50%, but it is unknown how many patients presented cardiovascular comorbidities (hypertension ,diabetes, valvular heart disease, coronary artery disease, LV hypertrophy....) that might contribute to a decreased LV or RV GLS

Response from Author:

We have added additional sensitivity analysis related to cardiovascular comorbidities on its influence toward LV-GLS and RV-GLS result.

Moreover, in order to evaluate whether cardiovascular comorbidities could have contribution to a decreased LV-GLS, we excluded studies that include patients with hypertension, diabetes mellitus, and left ventricular hypertrophy. Further exclusion of three studies[14,44,57] with hypertension and diabetes mellitus patients showed similar result (MD:-1.55;95%CI,-2.37 – -0.73,p=0.0002; I2=73%,p=0.0003) while exclusion of a study[43] with left ventricular hypertrophy patients also showed simillar results (MD:-1.59;95%CI,-2.34 – -0.84,p<0.00001; I2=70%,p=0.0002). Other comorbidities such as coronary artery disease and valvular heart disease were excluded by all studies and therefore these variable did not influence current result. [PAGE 12]

Moreover, in order to evaluate whether cardiovascular comorbidities could have contribution to a decreased RV-GLS, we excluded studies that include patients with hypertension, diabetes mellitus, and left ventricular hypertrophy. Further exclusion of two studies[14,57] with hypertension and diabetes mellitus patients showed similar result (MD:-2.09;95%CI,-3.05 – -1.13,p<0.0001; I2=0%,p=0.78). Among these studies, there were no study evaluating RV-GLS which include patients with left ventricular hypertrophy. Other comorbidities such as coronary artery disease and valvular heart disease were excluded by all studies and therefore these variable did not influence current result. [PAGE 13]

Response from Reviewer:

-It is not clear which was the relationship between LVGLS and LV ejection fraction/MAPSE, or between RVGLS and TAPSE/FAC; did you analyse this point? Was there a difference between patients with bordeline LVEF and normal/high LVEF?

Response from Author:

We have added additional analysis comparing TAPSE and FAC between groups. Additional figure are added in suplementary files. However, there was no study reporting MAPSE and thus evaluation are limited.

Other Parameters

We found no significant difference of tricuspid annular plane systolic excursion (TAPSE) between cirrhotic versus non-cirrhotic patients based on all available data which was reported by 6 studies (MD:0.30;95%CI,-0.25 – 0.85, p=0.29; I2=0%,p=0.59) (S6 Fig).[14,15,44,45,47,57] Similarly, all 4 studies[14,15,45,57] reporting right ventricular fractional area change (RVFAC) between cirrhotic and non-cirrhotic patients showed no significant difference (MD:-0.56;95%CI,-1.85 – 0.73, p=0.40; I2=0%,p=0.44) (S7 Fig). [PAGE 14]

Response from Reviewer:

-Besides an analysis on compensated/decompensated liver cirrhosis, were there differences according to ongoing medical therapy (diuretics? beta-blockers? ACE/ARB? MRA?) or to the presence of heart failure?

Response from Author:

We have added additional sensitivity analysis to evaluate the influence of heart failure (diastolic dysfunction, abnormal NT-proBNP result) and ongoing medical therapy that may affect the LV-GLS result. Moreover, we have update Table 1 by adding information regarding ongoing medical therapy and NT-proBNP. No study reporting the use of MRA in the population included and thus these variable was unable to be evaluated.

Additionaly, exclusion of two studies[15,47] which include patients with diastolic dysfunction revealed a similar result indicating a significant reduction of LV-GLS in cirrhotic patients (MD:-1.85;95%CI,-2.57 – -1.33,p<0.00001; I2=63%,p=0.003). In order to evaluate whether ongoing medical therapy (diuretics, beta-blockers, angiotensin converting enzyme inhibitor – ACE-I/angiotensin receptor blocker – ARB, calcium channel blocker – CCB) could influence LV-GLS, we exclude 3 studies[10,14,43] that report these medications in cirrhotic patients and the result showed that there is still a significant reduction of LV-GLS in cirrhotic patients (MD:-1.69;95%CI,-2.59 – -0.79,p=0.0002; I2=76%,p<0.0001). The effect of diuretic treatment that can be seen when a study[10] was excluded from the analysis also resulting in a similar result (MD:-1.72;95%CI,-2.40 – -1.04,p<0.00001; I2=71%,p=0.0002). Excluding a study[43] to evaluate the influence of beta-blocker treatment on the result also showing similar finding (MD:-1.59;95%CI,-2.34 – -0.84,p<0.0001; I2=70%,p=0.0002). [PAGE 12]

The effect of diastolic dysfunction on RV-GLS results is unable to be evaluated because limited data availble on the remaining study. The effect of ongoing medical therapy (diuretics, beta-blockers, ACE-I/ARB, CCB) on RV-GLS result was evaluated by excluding a study[14] which still showing a significant reduction of RV-GLS in cirrhotic patients (MD:-1.97;95%CI,-2.90 – -1.04,p<0.0001; I2=0%,p=0.57). [PAGE 13]

---

## [Editor Report · Decision Letter 1]

21 Mar 2022

PONE-D-21-37272R1Evaluation of Subclinical Ventricular Systolic Dysfunction Assessed Using Global Longitudinal Strain in Liver Cirrhosis: A Systematic Review and Meta-Analysis with Trial Sequential AnalysisPLOS ONE

Dear Dr. Ridjab,

Thank you for submitting your manuscript to PLOS ONE. After careful consideration, we feel that it has merit but does not fully meet PLOS ONE’s publication criteria as it currently stands. Therefore, we invite you to submit a revised version of the manuscript that addresses the points raised during the review process.

ACADEMIC EDITOR: All issues raised by expert reviewers are required.

We look forward to receiving your revised manuscript.

Kind regards,

Vincenzo Lionetti, M.D., PhD

Academic Editor

PLOS ONE

---

## [Author Response · Author response to Decision Letter 1]

22 Mar 2022

REVISION#1

Academic Editor

Response from Academic Editor:

All issues raised by expert reviewers are required. Moreover, the authors should revise statistical approach in order to better support the conclusions.

Response from Author:

We have revise our statistical approach. The reporting of the trial sequential analysis has been improved by defining the mean difference a priori, before the trial sequential analysis run while also correcting the heterogeneity to reach a low heterogeneity (at least 25%). With this approach we report a more robust result. This was further described in the results section (Trial Sequential Analysis) and indicated in Figure S10 and Figure S11. This means that all result from meta analysis have shown a conclusive evidence with a low risk of a false positive result. 

We calculated the information size required based on α = 5% significance level, and β = 20% (80% power) with an anticipated LV-GLS change of -1.43% and RV-GLS change of -1.95% in favor of cirrhotic groups. Heterogeneity was corrected to reach I2 of at least 25% (low heterogeneity). [PAGE 4]

TSA result for LV-GLS and RV-GLS are shown in S10 Fig and S11 Fig, respectively. Both TSA showed that these studies cross trial monitoring boundary and confirm a lower absolute GLS in favor of cirrhotic group. This indicate that result of GLS in both ventricle have low risk of false positive results. After conducting TSA with sensitivity analysis that has been done previously in Fig 4 for LV-GLS and Figure S4 for RV-GLS, a further TSA is then generated for LV-GLS (S12 Fig) and RV-GLS (S13 Fig). From these TSA, the result is also conclusive and have low risk of false positive results. [PAGE 12]

The conclusion have been better improved in accordance with the improved results by using a revised statistical approach

Despite the clinical and methodological heterogeneity, TSA revealed that current meta analysis have shown firm evidences that cirrhotic patients have a lower GLS when compared with controls. Future studies should provide a further analysis to elucidate whether there will be a significant difference of GLS in cirrhotic patients with different age, sex, cirrhosis severity and etiologies. [PAGE 16]

Journal Requirements

Response from Journal:

Response from Author:

We have followed all guidance accordingly

Response from Journal:

Response from Author:

We have update Data Availability information. We have attached all of our raw data (File Name: RAW DATA_PLOS ONE). No ethical or legal restriction in sharing our data.

Response from Journal:

Response from Author:

We have added captions for our Supporting Information files at the end of manuscript.

Reviewers' Comments

Reviewer's Responses to Questions

Comments to the Author:

1. Is the manuscript technically sound, and do the data support the conclusions?

Reviewer #1: Yes

Reviewer #2: Yes

Response from Author:

We thank all reviewers that have provided responses indicating that our manuscript is technically sound and that the data have support the conclusion

Comments to the Author:

2. Has the statistical analysis been performed appropriately and rigorously?

Reviewer #1: I Don't Know

Reviewer #2: I Don't Know

Response from Author:

In reporting our systematic review, we followed the Preferred Reporting Items for Systematic reviews and Meta-Analyses (PRISMA) 2020 statement guideline. (Reference Number 18)

We comply with Conducting Systematic Reviews and Meta-Analyses of Observational Studies of Etiology (COSMOS-E) guidance to undertake this systematic review and meta-analysis. (Reference Number 19)

Report of statistical heterogeneity and publication bias was in accordance with the Cochrane Handbook for Systematic Reviews of Interventions version 6.1. (Reference Number 22)

Sensitivity analysis was in accordance with a guideline from “Doing a Meta-Analysis with R: A Hands on Guide” (Reference Number 30)

Trial Sequential Analysis was performed in accordance with User manual for trial sequential analysis (TSA). (Reference Number 38)

Methodological quality evaluation was in accordance with The Newcastle-Ottawa Scale (NOS) for assessing the quality of nonrandomised studies in meta-analyses. (Reference Number 40)

All statistical analysis was performed using Review Manager 5.3.5 software (Copenhagen: The Nordic Cochrane Centre, The Cochrane Collaboration) and JASP Version 0.14.1

We believe by following all of these guidelines, our statistical analysis has been performed appropriately and rigorously. Moreover, we have attached all of our raw data (File Name: RAW DATA_PLOS ONE) in which all of these data could be utilized to replicate the reported study findings in their entirety. 

Comments to the Author:

3. Have the authors made all data underlying the findings in their manuscript fully available?

Reviewer #1: Yes

Reviewer #2: Yes

Response from Author:

We thank all reviewers that have provided responses indicating that the authors have made all data underlying the findings in manuscript fully available

Comments to the Author:

4. Is the manuscript presented in an intelligible fashion and written in standard English?

Reviewer #1: Yes

Reviewer #2: Yes

Response from Author:

We thank all reviewers that have provided responses indicating that the manuscript is presented in an intelligible fashion and written in standard English.

Response from Reviewer:

Reviewer #1: Dear Authors, Thank you for the opportunity to read your work. After read it with attention the results seems to me sound. I'm not totally able to assess the statistical analysis, but it appear to me to be right. Please spent time to siimplify the text that is now too technical. I will read your revision version with pleasure.

Response from Authors:

We have simplify the text particularly the discussion part in order to be more readable. Several grammatical errors have also been corrected. [PAGE 2-16]

Response from Reviewer:

Reviewer #2: The Authors performed a detailed metanalysis showing a significant reduction of LV-GLS and RV-GLS in cirrhotic patients. Overall, they performed an extensive search and analysis from as many as 5347 studies, yielding 20 studies (1738 pts; 2 studies with CMR, 18 with echo). Please find some comments:

-All patients presented by definition a LVEF >50%, but it is unknown how many patients presented cardiovascular comorbidities (hypertension ,diabetes, valvular heart disease, coronary artery disease, LV hypertrophy....) that might contribute to a decreased LV or RV GLS

Response from Author:

We have added additional sensitivity analysis related to cardiovascular comorbidities on its influence toward LV-GLS and RV-GLS result.

Moreover, in order to evaluate whether cardiovascular comorbidities could have contribution to a decreased LV-GLS, we excluded studies that include patients with hypertension, diabetes mellitus, and left ventricular hypertrophy. Further exclusion of three studies[14,44,57] with hypertension and diabetes mellitus patients showed similar result (MD:-1.55;95%CI,-2.37 – -0.73,p=0.0002; I2=73%,p=0.0003) while exclusion of a study[43] with left ventricular hypertrophy patients also showed simillar results (MD:-1.59;95%CI,-2.34 – -0.84,p<0.00001; I2=70%,p=0.0002). Other comorbidities such as coronary artery disease and valvular heart disease were excluded by all studies and therefore these variable did not influence current result. [PAGE 12]

Moreover, in order to evaluate whether cardiovascular comorbidities could have contribution to a decreased RV-GLS, we excluded studies that include patients with hypertension, diabetes mellitus, and left ventricular hypertrophy. Further exclusion of two studies[14,57] with hypertension and diabetes mellitus patients showed similar result (MD:-2.09;95%CI,-3.05 – -1.13,p<0.0001; I2=0%,p=0.78). Among these studies, there were no study evaluating RV-GLS which include patients with left ventricular hypertrophy. Other comorbidities such as coronary artery disease and valvular heart disease were excluded by all studies and therefore these variable did not influence current result. [PAGE 13]

Response from Reviewer:

-It is not clear which was the relationship between LVGLS and LV ejection fraction/MAPSE, or between RVGLS and TAPSE/FAC; did you analyse this point? Was there a difference between patients with bordeline LVEF and normal/high LVEF?

Response from Author:

We have added additional analysis comparing TAPSE and FAC between groups. Additional figure are added in suplementary files. However, there was no study reporting MAPSE and thus evaluation are limited.

Other Parameters

We found no significant difference of tricuspid annular plane systolic excursion (TAPSE) between cirrhotic versus non-cirrhotic patients based on all available data which was reported by 6 studies (MD:0.30;95%CI,-0.25 – 0.85, p=0.29; I2=0%,p=0.59) (S6 Fig).[14,15,44,45,47,57] Similarly, all 4 studies[14,15,45,57] reporting right ventricular fractional area change (RVFAC) between cirrhotic and non-cirrhotic patients showed no significant difference (MD:-0.56;95%CI,-1.85 – 0.73, p=0.40; I2=0%,p=0.44) (S7 Fig). [PAGE 14]

Response from Reviewer:

-Besides an analysis on compensated/decompensated liver cirrhosis, were there differences according to ongoing medical therapy (diuretics? beta-blockers? ACE/ARB? MRA?) or to the presence of heart failure?

Response from Author:

We have added additional sensitivity analysis to evaluate the influence of heart failure (diastolic dysfunction, abnormal NT-proBNP result) and ongoing medical therapy that may affect the LV-GLS result. Moreover, we have update Table 1 by adding information regarding ongoing medical therapy and NT-proBNP. No study reporting the use of MRA in the population included and thus these variable was unable to be evaluated.

Additionaly, exclusion of two studies[15,47] which include patients with diastolic dysfunction revealed a similar result indicating a significant reduction of LV-GLS in cirrhotic patients (MD:-1.85;95%CI,-2.57 – -1.33,p<0.00001; I2=63%,p=0.003). In order to evaluate whether ongoing medical therapy (diuretics, beta-blockers, angiotensin converting enzyme inhibitor – ACE-I/angiotensin receptor blocker – ARB, calcium channel blocker – CCB) could influence LV-GLS, we exclude 3 studies[10,14,43] that report these medications in cirrhotic patients and the result showed that there is still a significant reduction of LV-GLS in cirrhotic patients (MD:-1.69;95%CI,-2.59 – -0.79,p=0.0002; I2=76%,p<0.0001). The effect of diuretic treatment that can be seen when a study[10] was excluded from the analysis also resulting in a similar result (MD:-1.72;95%CI,-2.40 – -1.04,p<0.00001; I2=71%,p=0.0002). Excluding a study[43] to evaluate the influence of beta-blocker treatment on the result also showing similar finding (MD:-1.59;95%CI,-2.34 – -0.84,p<0.0001; I2=70%,p=0.0002). [PAGE 12]

The effect of diastolic dysfunction on RV-GLS results is unable to be evaluated because limited data availble on the remaining study. The effect of ongoing medical therapy (diuretics, beta-blockers, ACE-I/ARB, CCB) on RV-GLS result was evaluated by excluding a study[14] which still showing a significant reduction of RV-GLS in cirrhotic patients (MD:-1.97;95%CI,-2.90 – -1.04,p<0.0001; I2=0%,p=0.57). [PAGE 13]

Comments to the Author:

6. PLOS authors have the option to publish the peer review history of their article (what does this mean?). If published, this will include your full peer review and any attached files.

Do you want your identity to be public for this peer review? For information about this choice, including consent withdrawal, please see our Privacy Policy.

Reviewer #1: Yes: Luigi Vetrugno

Reviewer #2: No

Response from Author:

We thank all the reviewers for providing comments to our manuscript. We have provided feedback to all of the reviewers comments point by point.

REVISION #2

Academic Editor

Response from Academic Editor:

All issues raised by expert reviewers are required.

Response from Authors:

We have revised our attachment file in “Response to Reviewers” section in order to provide a more complete response to all of the statements given in the previous Revision round. We can not find any additional issues raised by reviewers in this second revision within the letter sent on March 21 2022 01:53PM. In this second round revision, we also did not find any action link (“View Attachment”) as indicated by the journal if reviewer comments were submitted as an attachment file. Please let us know if there is an error that make us unable to find additional issues from reviewers in the letter provided on March 21 2022 01:53PM.

Response from Journal:

Response from Author:

We have used PACE to ensure that figures meet PLOS requirments.

---

## [Decision Letter · Decision Letter 2]

11 Apr 2022

PONE-D-21-37272R2Evaluation of Subclinical Ventricular Systolic Dysfunction Assessed Using Global Longitudinal Strain in Liver Cirrhosis: A Systematic Review and Meta-Analysis with Trial Sequential AnalysisPLOS ONE

Dear Dr. Ridjab,

Thank you for submitting your manuscript to PLOS ONE. After careful consideration, we feel that it has merit but does not fully meet PLOS ONE’s publication criteria as it currently stands. Therefore, we invite you to submit a revised version of the manuscript that addresses the points raised during the review process.

ACADEMIC EDITOR: Statistical revision was performed by expert in addition to selected reviewers in order to improve the quality of the manuscript. Major issues need to be adequately solved by the authors as reported by the third reviewer. These issues are required.

We look forward to receiving your revised manuscript.

Kind regards,

Vincenzo Lionetti, M.D., PhD

Academic Editor

PLOS ONE

Reviewers' comments:

Reviewer's Responses to Questions

**Comments to the Author**

1. If the authors have adequately addressed your comments raised in a previous round of review and you feel that this manuscript is now acceptable for publication, you may indicate that here to bypass the “Comments to the Author” section, enter your conflict of interest statement in the “Confidential to Editor” section, and submit your "Accept" recommendation.

Reviewer #1: All comments have been addressed

Reviewer #2: All comments have been addressed

Reviewer #3: (No Response)

2. Is the manuscript technically sound, and do the data support the conclusions?

Reviewer #1: Yes

Reviewer #2: Yes

Reviewer #3: Partly

3. Has the statistical analysis been performed appropriately and rigorously? 

Reviewer #1: Yes

Reviewer #2: Yes

Reviewer #3: No

4. Have the authors made all data underlying the findings in their manuscript fully available?

Reviewer #1: Yes

Reviewer #2: Yes

Reviewer #3: Yes

5. Is the manuscript presented in an intelligible fashion and written in standard English?

Reviewer #1: Yes

Reviewer #2: Yes

Reviewer #3: No

6. Review Comments to the Author

Reviewer #1: The authors have address the issue raise by the reviewers. Now the manuscript appear clear and the statistical part is revised.

Best Regards

Reviewer #2: (No Response)

Reviewer #3: The systematic review appears to be rather thorough. The meta-analysis is carried out routinely as one would expect. The publication bias and sensitivity analysis are performed as one would expect. The results appear to follow from the analyses.

The confusion comes in the inclusion of the studies and their exact description from a statistical perspective. Table 1 is not much help. Although described in the text, there are 12 case control and cross sectional studies and 8 prospective studies. The prospective cohort are not well described. Exactly how many studies had a direct randomized comparison? In order to make valid statistical comparisons overall, these should be looked at separately.

The addition of the TSA adds further confusion. It’s exact purpose is not clearly stated. How does it add statistically to the strength of the meta-analytic results?

As seen in Figures 2 to 4. The heterogeneity is rather high especially in Figures 2 and 3. The investigators do not seem to have investigated the true causes of heterogeneity and explain such a high value of the I-square statistic. Also, the overall p-value of the p=0.04 is borderline significant. The overall fail safe number of publications to give assurance to these values should have been calculated.

The authors overwhelm the reader with many analyses and sub analyses and one can lose their focus with such a confusing statistical presentation.

7. PLOS authors have the option to publish the peer review history of their article (what does this mean?). If published, this will include your full peer review and any attached files.

Reviewer #1: **Yes: **Luigi Vetrugno

Reviewer #2: No

Reviewer #3: No

---

## [Author Response · Author response to Decision Letter 2]

22 May 2022

REVISION #3

Academic Editor

Response from Academic Editor:

Statistical revision was performed by expert in addition to selected reviewers in order to improve the quality of the manuscript. Major issues need to be adequately solved by the authors as reported by the third reviewer. These issues are required.

Response from Author:

We have revise our statistical approach:

1. We have done meta regression in order to investigate the true causes of heterogeneity and provide explaination of factor that associated with high I-square value. (PAGE 4, LINE 129; PAGE 16, LINE 370; PAGE 17, LINE 397)

2. We have removed TSA to reduce further confusion due to many statistical presentation (PAGE 4, LINE 148)

3. We have added meta analysis report based on study design separately (PAGE 14, LINE 318)

4. A fail safe N analysis have been added to give assurance to the borderline significant statistic. Result shows a robust finding. (PAGE 4, LINE 124; PAGE 16, LINE 363; PAGE 17, LINE 402)

Reviewers' Comments

Reviewer's Responses to Questions

Comments to the Author:

1. If the authors have adequately addressed your comments raised in a previous round of review and you feel that this manuscript is now acceptable for publication, you may indicate that here to bypass the “Comments to the Author” section, enter your conflict of interest statement in the “Confidential to Editor” section, and submit your "Accept" recommendation.

Reviewer #1: All comments have been addressed

Reviewer #2: All comments have been addressed

Reviewer #3: (No Response)

Response from Author:

We thank all reviewers that have provided responses indicating that our manuscript have addressed all comments

Comments to the Author:

2. Is the manuscript technically sound, and do the data support the conclusions?

Reviewer #1: Yes

Reviewer #2: Yes

Reviewer #3: Partly

Response from Author:

We thank all reviewers that have provided responses indicating that our manuscript is technically sound and the data support the conclusions. Additionaly, we have revised our manuscript following Reviewer #3 comments.

Comments to the Author:

3. Has the statistical analysis been performed appropriately and rigorously?

Reviewer #1: Yes

Reviewer #2: Yes

Reviewer #3: No

Response from Author:

We thank all reviewers that have provided responses indicating that the statistical analysis has been performed appropriately and rigorously. Additionaly, we have revised our manuscript following Reviewer #3 comments.

Comments to the Author:

4. Have the authors made all data underlying the findings in their manuscript fully available?

Reviewer #1: Yes

Reviewer #2: Yes

Reviewer #3: Yes

Response from Author:

We thank all reviewers that have provided responses indicating that the authors have made all data underlying the findings in manuscript fully available

Comments to the Author:

5. Is the manuscript presented in an intelligible fashion and written in standard English?

Reviewer #1: Yes

Reviewer #2: Yes

Reviewer #3: No

Response from Author:

Following Reviewer #3 response, we have sent our manuscript to Editage English Editing Service to improve our manuscript writing and comply with the standard for English scientific writing. We have uploaded the certificate of editing with the file name “Certificate_of_editing-DEIDJ_1”. 

Response from Reviewer:

6. Review Comments to the Author

Reviewer #1: The authors have address the issue raise by the reviewers. Now the manuscript appear clear and the statistical part is revised. Best Regards.

Response from Author:

We thank Reviewer #1 for all the comments indicating that our manuscript appear clear and the statistical part is revised.

Response from Reviewer:

Reviewer #2: (No Response)

Response from Author:

We thank Reviewer #2 for all the comments in the previous revision.

Response from Reviewer:

Reviewer #3: The systematic review appears to be rather thorough. The meta-analysis is carried out routinely as one would expect. The publication bias and sensitivity analysis are performed as one would expect. The results appear to follow from the analyses.

Response from Author:

We thank Reviewer #3 that have provided responses indicating that our manuscript appears to be rather thorough, the meta analysis was carried out routinely as expected, publication bias and sensitivity analysis performed as expected, and results appear to follow from the analyses.

Response from Reviewer:

Reviewer #3: The confusion comes in the inclusion of the studies and their exact description from a statistical perspective. 

Response from Author:

For studies inclusion, the description of all findings have been detailed based on study design. (PAGE 5, LINE 166; PAGE 12, LINE 206). For statistical revision we have performed the following:

1. We have done meta regression in order to investigate the true causes of heterogeneity and provide explaination of factor that associated with high I-square value. (PAGE 4, LINE 129; PAGE 16, LINE 370; PAGE 17, LINE 397)

2. We have removed TSA to reduce further confusion due to many statistical presentation (PAGE 4, LINE 148)

3. We have added meta analysis report based on study design separately (PAGE 14, LINE 318)

4. A fail safe N analysis have been added to give assurance to the borderline significant statistic. Result shows a robust finding. (PAGE 4, LINE 124; PAGE 16, LINE 363; PAGE 17, LINE 402)

Response from Reviewer:

Table 1 is not much help. 

Response from Author:

Table 1 have been simplified. 

Response from Reviewer:

Although described in the text, there are 12 case control and cross sectional studies and 8 prospective studies. 

The description of all findings have been detailed based on study design. (PAGE 5, LINE 166 – LINE 181)

“This systematic review identified... Other comorbidities such as dyslipidemia and pulmonary artery hypertension were reported in one cross-sectional [46] and one case-control study[51], respectively.” 

The description of all findings have been detailed based on study design. (PAGE 12, LINE 206 – LINE 216)

“For left ventricular assessment, there were two cross-sectional… included patients with a borderline EF (50-55%).”

Response from Reviewer:

The prospective cohort are not well described. 

Response from Author:

We have extensively added description of all prospective cohort studies. (PAGE 5, LINE 182 – LINE 200)

“All five prospective cohorts had different durations… The mean MELD score was not reported.”

Response from Reviewer:

Exactly how many studies had a direct randomized comparison? In order to make valid statistical comparisons overall, these should be looked at separately.

Response from Author:

We have added new statement that our search strategy did not identify any randomized controlled trial. (PAGE 5, LINE 168)

 “Our search strategy did not identify any randomized controlled trials.”

Response from Reviewer:

The addition of the TSA adds further confusion. It’s exact purpose is not clearly stated. How does it add statistically to the strength of the meta-analytic results?

Response from Author:

We have removed TSA to reduce further confusion due to many statistical presentation. (PAGE 4, LINE 148)

Response from Reviewer:

As seen in Figures 2 to 4. The heterogeneity is rather high especially in Figures 2 and 3. The investigators do not seem to have investigated the true causes of heterogeneity and explain such a high value of the I-square statistic. 

Response from Author:

We have done meta regression in order to investigate the true causes of heterogeneity and provide explaination of factor that associated with high I-square value. (METHODS, PAGE 4, LINE 129 – LINE 135)

“We performed a random-effects (method of moments) meta-regression… baseline LVEF in the group with cirrhosis, and methodological quality of the study.”

We have done meta regression in order to investigate the true causes of heterogeneity and provide explaination of factor that associated with high I-square value. (RESULT, PAGE 16, LINE 370 – LINE 377)

“Meta-regression analysis performed to explain variations in the reduction… All data results of the meta-regression are shown in S12-S20 Table.”

We have done meta regression in order to investigate the true causes of heterogeneity and provide explaination of factor that associated with high I-square value. (DISCUSSION, PAGE 17, LINE 397 – LINE 402)

“We performed meta-regression to investigate the true cause of heterogeneity influencing… a higher GLS was linked to more severe liver disease.”

Response from Reviewer:

Also, the overall p-value of the p=0.04 is borderline significant. The overall fail safe number of publications to give assurance to these values should have been calculated.

Response from Author:

A fail safe N analysis have been added to give assurance to the borderline significant statistic. Result shows a robust finding. (METHODS, PAGE 4, LINE 124 – LINE 128)

“The overall fail-safe number of publications… five times higher than the number of studies included, plus 10”

A fail safe N analysis have been added to give assurance to the borderline significant statistic. Result shows a robust finding. (RESULTS, PAGE 16, LINE 363 – LINE 367)

“Based on all the meta-analyses performed, Rosenthal’s fail-safe number… resulted in a fail-safe number of 317.”

A fail safe N analysis have been added to give assurance to the borderline significant statistic. Result shows a robust finding. (DISCUSSION, PAGE 17, LINE 402 – LINE 405)

“Additionally, although some of our meta-analyses… exceeded the minimum limit, as proposed by Rosenthal et al.”

Response from Reviewer:

The authors overwhelm the reader with many analyses and sub analyses and one can lose their focus with such a confusing statistical presentation.

Response from Author:

We have removed TSA to reduce further confusion due to many statistical presentation (PAGE 4, LINE 148).

Comments to the Author:

7. PLOS authors have the option to publish the peer review history of their article (what does this mean?). If published, this will include your full peer review and any attached files.

Do you want your identity to be public for this peer review? For information about this choice, including consent withdrawal, please see our Privacy Policy.

Reviewer #1: Yes: Luigi Vetrugno

Reviewer #2: No

Reviewer #3: No

Response from Author:

We thank all the reviewers for providing comments to our manuscript. We have provided feedback to all of the reviewers comments point by point.

---

## [Decision Letter · Decision Letter 3]

26 May 2022

Evaluation of subclinical ventricular systolic dysfunction assessed using global longitudinal strain in liver cirrhosis: a systematic review, meta-analysis, and meta-regression

PONE-D-21-37272R3

Dear Dr. Ridjab,

We’re pleased to inform you that your manuscript has been judged scientifically suitable for publication and will be formally accepted for publication once it meets all outstanding technical requirements.

Kind regards,

Vincenzo Lionetti, M.D., PhD

Academic Editor

PLOS ONE

Additional Editor Comments (optional):

Reviewers' comments:

Reviewer's Responses to Questions

**Comments to the Author**

1. If the authors have adequately addressed your comments raised in a previous round of review and you feel that this manuscript is now acceptable for publication, you may indicate that here to bypass the “Comments to the Author” section, enter your conflict of interest statement in the “Confidential to Editor” section, and submit your "Accept" recommendation.

Reviewer #1: All comments have been addressed

Reviewer #2: All comments have been addressed

Reviewer #3: All comments have been addressed

2. Is the manuscript technically sound, and do the data support the conclusions?

Reviewer #1: Yes

Reviewer #2: Yes

Reviewer #3: (No Response)

3. Has the statistical analysis been performed appropriately and rigorously? 

Reviewer #1: Yes

Reviewer #2: I Don't Know

Reviewer #3: (No Response)

4. Have the authors made all data underlying the findings in their manuscript fully available?

Reviewer #1: Yes

Reviewer #2: Yes

Reviewer #3: (No Response)

5. Is the manuscript presented in an intelligible fashion and written in standard English?

Reviewer #1: Yes

Reviewer #2: Yes

Reviewer #3: (No Response)

6. Review Comments to the Author

Reviewer #1: Thank you for addressing my comments.

This is R3 and I have not further question to you.

Best Regards

Reviewer #2: (No Response)

Reviewer #3: (No Response)

7. PLOS authors have the option to publish the peer review history of their article (what does this mean?). If published, this will include your full peer review and any attached files.

Reviewer #1: **Yes: **Luigi Vetrugno

Reviewer #2: No

Reviewer #3: No

---

## [Editor Report · Acceptance letter]

30 May 2022

PONE-D-21-37272R3 

Evaluation of subclinical ventricular systolic dysfunction assessed using global longitudinal strain in liver cirrhosis: a systematic review, meta-analysis, and meta-regression 

Dear Dr. Ridjab:

I'm pleased to inform you that your manuscript has been deemed suitable for publication in PLOS ONE. Congratulations! Your manuscript is now with our production department. 

Kind regards, 

on behalf of

Prof. Vincenzo Lionetti 

Academic Editor

PLOS ONE